# Engineering the haemogenic niche mitigates endogenous inhibitory signals and controls pluripotent stem cell-derived blood emergence

Nafees Rahman[1,2], Patrick M. Brauer[3], Lilian Ho[4], Tatiana Usenko[2], Mukul Tewary[2,8], Juan Carlos Zúñiga-Pflücker[3,5,6] & Peter W. Zandstra[1,2,6,7,8]

Efforts to recapitulate haematopoiesis, a process guided by spatial and temporal inductive signals, to generate haematopoietic progenitors from human pluripotent stem cells (hPSCs) have focused primarily on exogenous signalling pathway activation or inhibition. Here we show haemogenic niches can be engineered using microfabrication strategies by micropatterning hPSC-derived haemogenic endothelial (HE) cells into spatially-organized, size-controlled colonies. CD34 + VECAD + HE cells were generated with multi-lineage potential in serum-free conditions and cultured as size-specific haemogenic niches that displayed enhanced blood cell induction over non-micropatterned cultures. Intra-colony analysis revealed radial organization of CD34 and VECAD expression levels, with CD45 + blood cells emerging primarily from the colony centroid area. We identify the induced interferon gamma protein (IP-10)/p-38 MAPK signalling pathway as the mechanism for haematopoietic inhibition in our culture system. Our results highlight the role of spatial organization in hPSC-derived blood generation, and provide a quantitative platform for interrogating molecular pathways that regulate human haematopoiesis.

[1] Department of Chemical Engineering and Applied Chemistry, University of Toronto, Toronto, Ontario, Canada M5S 3ES. [2] Institute of Biomaterials and Biomedical Engineering, University of Toronto, Toronto, Ontario, Canada M5S 3G9. [3] Biological Sciences, Sunnybrook Research Institute, Toronto, Ontario, Canada M4N 3M5. [4] Life Sciences (Biochemistry), University of Waterloo, Waterloo, Ontario, Canada N2L 3G1. [5] Department of Immunology, University of Toronto, Toronto, Ontario, Canada M5S 1A8. [6] Medicine by Design, a Canada First Research Excellence Program at the University of Toronto, Toronto, Ontario, Canada M5S 3G9. [7] Terrence Donnelly Centre for Cellular & Biomolecular Research, University of Toronto, Toronto, Ontario, Canada M5S 3E1. [8] Collaborative Program in Developmental Biology, University of Toronto, Toronto, Ontario, Canada M5S 3E1. Correspondence and requests for materials should be addressed to P.W.Z. (email: peter.zandstra@utoronto.ca).

Human pluripotent stem cells (hPSCs) facilitate strategies to model human development and disease[1], and may serve as a renewable source of cells in a variety of cell therapy applications[2]. However, current protocols for the generation of hPSC-derived cells often fail to deliver mature, adult-like cells[3]. For example, the generation of definitive blood progenitors that behave similar to blood stem cells isolated from somatic tissue remains a challenge. Blood progenitor cells arise from haemogenic endothelium (HE) through a process termed endothelial-to-haematopoietic transition (EHT)[4,5]. This process occurs at specific locations during development and is spatially and temporally regulated by a balance of activating and inhibiting signals that is not completely understood. A well-studied region for mammalian blood cell emergence is the aorta-gonad-mesonephros (AGM). The first human definitive haematopoietic stem cells (HSCs) are spatially restricted to the ventral floor of the dorsal aorta[6]. In the AGM, paracrine signals from tissues ventral to the dorsal aorta (such as the mesenchyme, primitive gut and sympathetic nervous system) promote haematopoiesis, while tissues dorsal to the dorsal aorta (such as the neural tube and notochord) suppress blood formation[7–10]. These observations support our hypothesis that exerting spatial control over the local microenvironment of hPSC-derived HE will modulate blood cell yields and provide a platform to reveal organizing principles for this difficult-to-access developmental event.

Micropatterning has been used to spatially organize cells, allowing investigation of endogenous autocrine and paracrine signalling[11,12]. Using hPSCs, we have previously demonstrated that spatial control of endogenous BMP2 and GDF3 signalling can directly modulate pluripotency[11]. In addition, we have shown mouse embryonic stem cell colony size manipulation can control JAK-STAT activation, enabling subsequent transition towards epiblast stem cells[13,14]. Similarly, others have shown that geometric confinement of hPSCs can be used to recapitulate germ layer patterning. Herein we extend this approach to the micropatterning of hPSC-derived HE, and use this platform to control microenvironmental signals and spatial gradients during blood progenitor cell development.

We specifically report the serum-free generation of hPSC-derived HE cells capable of producing both myeloid and lymphoid progenitors. We explore key engineered niche parameters for their impact on blood cell development and identify conditions that enhance blood cell generation. We subsequently identify interferon gamma-induced protein (IP)-10 as an endogenous inhibitory factor for hPSC- and cord blood-derived blood induction. Furthermore, we use live cell imaging to visualize location-dependent human CD45+ cell emergence. Our results demonstrate the use of in vitro engineered cell niches to enhance PSC-derived blood cell development and provide a quantitative platform for interrogating molecular pathways that regulate this process.

## Results

**Serum-free generation of blood progenitor cells**. To study the role of spatial control on signalling and developmental dynamics during in vitro blood differentiation we developed a protocol for efficient production of HE cells from which to generate and isolate appropriate progenitor populations (Fig. 1a). Using RUNX1C-GFP HES3 cells, we used Aggrewell plates to generate 1,000 cell hPSC aggregates by forced aggregation[15], a size previously reported to be optimal for mesoderm differentiation[16,17], and cultured them under hypoxic conditions (5% $O_2$). Differentiating aggregates were transferred from Aggrewell plates to low-attachment six-well plates after 5 days (Supplementary Fig. 1b—left column) and placed on a shaker for the remaining 20 days of hypoxic culture. Using this

differentiation platform, we screened three serum-free differentiation protocols[18–20] for production of kinase insert domain receptor/tyrosine protein kinase kit/epithelial cadherin (KDR + CKIT − ECAD − )-expressing cells (Supplementary Fig. 1a), an early population that has been postulated to contain HE cells[21]. The protocol based on Ng et al.[22] yielded significant levels of KDR + CKIT − ECAD − cells in our hands and was used to generate hPSC-derived HE cells in all subsequent studies. The haematopoietic differentiation profile was phenotypically assessed by tracking protein expression of VECAD, CD34, CD43 and CD45 by flow cytometry. Expression of CD34 and VECAD peaked at day 8, whereas committed haematopoietic progenitors marked by CD43 and CD45 were detectable by day 10 and continued increasing until day 25 (Fig. 1b). Colony-forming cell (CFC) generation paralleled CD43 and CD45 expression and was significantly higher at day 25 ($P \leq 0.05$, $n = 3$, one-way analysis of variance (ANOVA) with post hoc Tukey test; Fig. 1c) compared to day 13 (see Supplementary Fig. 1b for representative colonies of myeloid lineage from hPSC-derived blood progenitor cells). Moreover, early CD43 expression marks differentiating cells capable of giving rise to erythroid progenitors (Ery-P) indicating these cells represent the primitive wave of embryonic haematopoiesis (Supplementary Fig. 2a)[23,24]. This protocol for haematopoietic differentiation was robustly transferable to H1 (Supplementary Fig. 1b), HES2 and H9 (Supplementary Fig. 2) cell lines. Our results demonstrate that hPSCs can faithfully recapitulate embryonic haematopoiesis in a growth factor-defined, serum-free differentiation protocol (see Fig. 1d for representative flow cytometry plots). We next sought to isolate an intermediate population that possess characteristics of HE cells to investigate spatial control of signalling during blood cell differentiation.

**Characterization of an hPSC-derived HE cell population**. To characterize HE populations capable of blood cell generation we used CD34 + VECAD + expression[6] to isolate cells from day 8 differentiating hPSC aggregates (see Fig. 2a for representative flow cytometry plots). We also assessed CD184 and CD73 expression[21,25] (Supplementary Fig. 2b), which have previously been shown to identify arterial and venous endothelium, respectively. To maximize the yield of day 8 CD34 + VECAD + cells, we performed an aggregate size screen (100 to 1,000 cells per aggregate) and determined the highest yield of CD34 + VECAD + cells per input hPSC, $0.32 \pm 0.015$ (mean ± s.e.m.), was obtained in cultures initiated with 500 cell aggregates ($P \leq 0.05$, $n = 3$, one-way ANOVA with post hoc Tukey test; Supplementary Fig. 1c). The CD34 + VECAD + HE cells generated using our protocol were composed of $34.5\% \pm 3.5\%$ (mean ± s.e.m.) CD73 − CD184 − cells, a substantially higher frequency than has been previously reported[25,26]. Lack of both markers can be used to distinguish HE-enriched for multi-lineage progenitors that can efficiently give rise to blood cells of myeloid and lymphoid lineages. Moreover, our serum-free differentiation protocol was robustly transferable across multiple hPSC lines, including the HES2 and H9 lines (Supplementary Fig. 2c). To functionally validate the HE phenotype of day 8 hPSC-derived cells, cell sorting was performed based on CD34 and VECAD expression (Fig. 2a). Positive fractions (CD34 + VECAD + ) were confirmed using immunocytochemistry and were capable of metabolizing DiI-acetylated-low-density lipoprotein (DiI-Ac-LDL) and stained exclusively for von Willebrand factor (vWF; Fig. 2b), consistent with characteristics of HE cells[5]. For comparative purposes, the impact of SB-431542 and CHIR99012 on HE induction in our serum-free differentiation protocol was measured[27]. The addition of SB-431542 led to a twofold increase in the frequency of CD34 + cells on day 6 compared to untreated control conditions, while the addition of

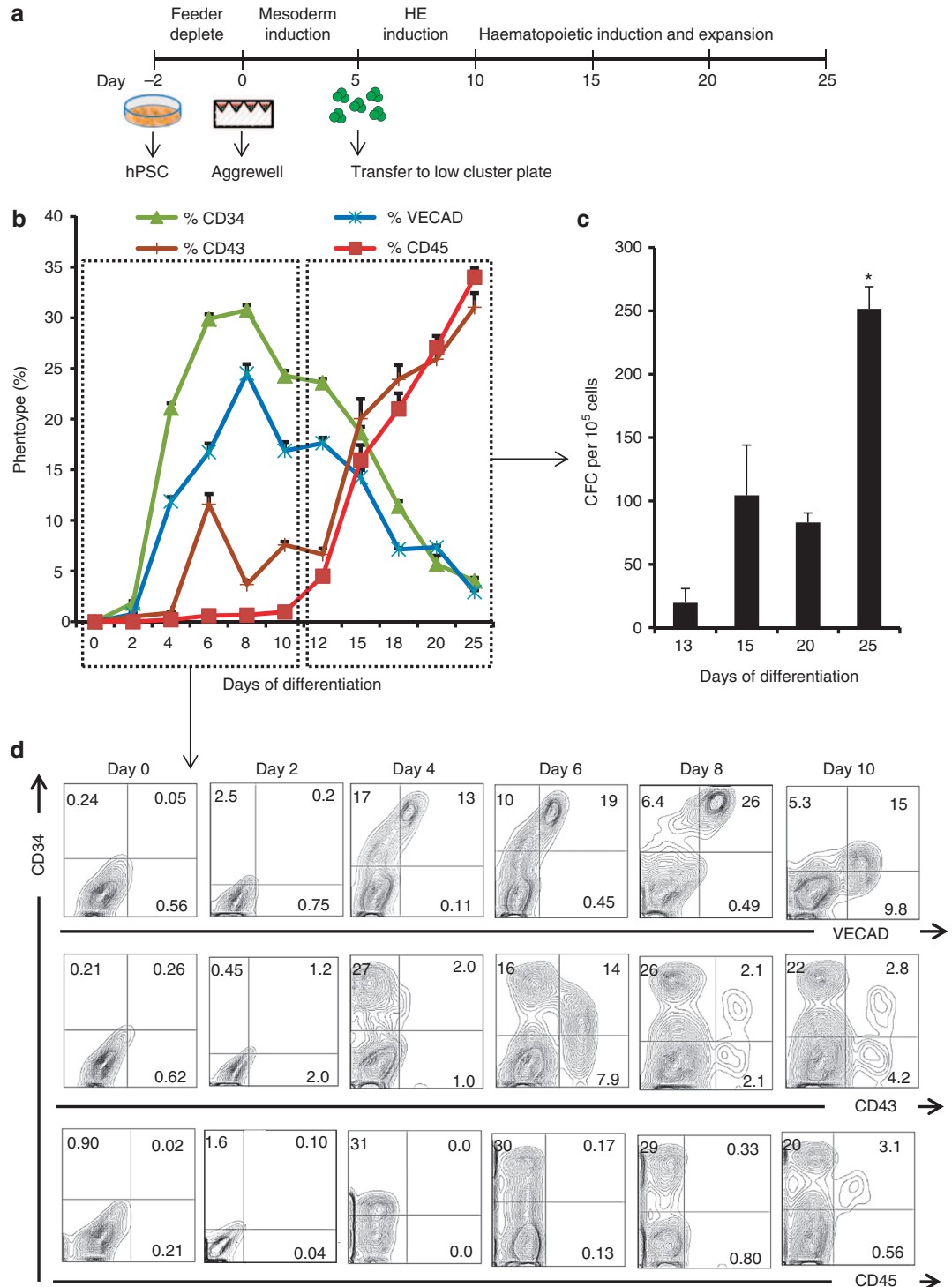

**Figure 1 | Characterization of hPSC differentiation to haematopoietic progenitor cells.** (**a**) Overview of serum-free differentiation protocol: size-controlled hPSC aggregates are formed in Aggrewell and differentiated for 25 days in serum-free medium. (**b**) Phenotype tracking of blood progenitor cells during differentiation, $n = 4$. (**c**) Enumeration of CFCs from differentiating day 11–25 hPSCs, $n = 3$. (**d**) Representative flow cytometry plots from day 0 to 10 depicting CD34, VECAD, CD43 and CD45 expression. Data are presented as mean ± s.e.m. Treatments displaying asterisks are significantly different from other groups ($P \leq 0.05$, one-way ANOVA *post hoc* Tukey test). See also Supplementary Fig. 1.

CHIR99012 did not significantly enhance CD34 expression compared to control treatments (Supplementary Fig. 2d). The CD34+ expression frequency observed in SB-431542-treated Aggrewell cultures was substantially higher than previously reported ($\sim 60\%$ versus 20%)[27], a result attributed to the

generation of uniform aggregates of optimal size. Furthermore, CD235a (a marker indicating progenitor cells primed for primitive haematopoiesis) was minimally expressed in our differentiation culture compared to SB-43152- and CHIR99012-supplemented treatments (Supplementary Fig. 3A). Collectively,

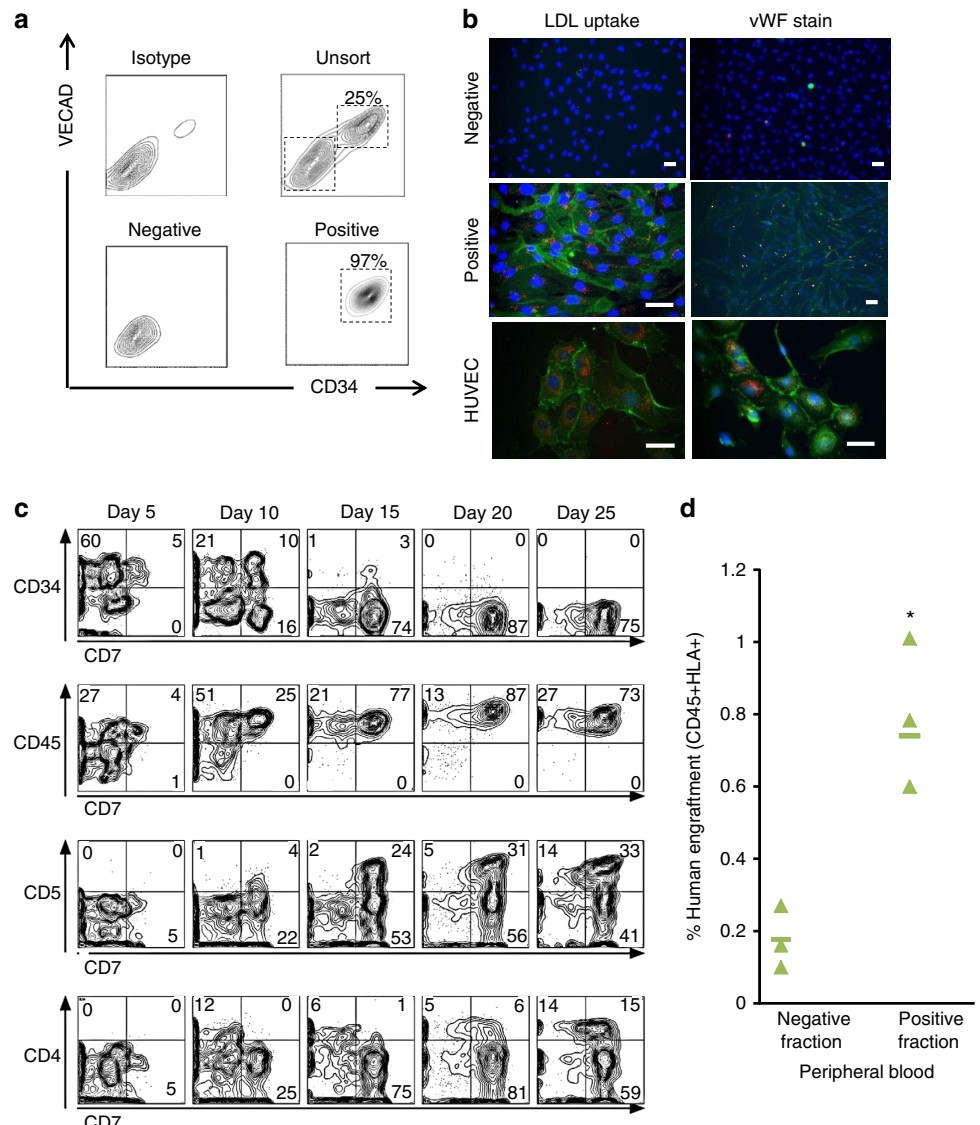

**Figure 2 | Day 8 hPSC-derived cells display HE characteristics and possess definitive blood potential.** (**a**) Representative day 8 flow cytometry plots showing isotype, unsorted, sorted negative fraction and sorted positive fractions. (**b**) Immunocytochemistry of day 8 CD34/VECAD fractions show DiI-Ac-LDL (red speckled) and vWF production (punctate red) only in positive cell fractions. Human umbilical vein endothelial cells (HUVECs) were used as positive controls. Positive fractions and HUVECs are concomitantly stained with VECAD (green—VECAD) and Hoechst (blue—Hoechst). Scale bar, 50 μm. (**c**) Kinetics of T-cell development of day 8 aggregate-derived hPSCs, enriched via magnetic-activated cell sorting, shows downregulation of CD34 expression and upregulation of CD45, CD7 and CD5 expression at indicated time points of co-culture with OP9-DL4 stroma, $n = 3$. Cord blood-derived cells were used as positive controls for stromal culture (not shown). (**d**) Phenotypic analysis for detection of human CD45 + HLA + expression in peripheral blood of NSG recipients 10 weeks post injection, $n = 3$. Treatments displaying asterisks are significantly different from other groups ($P \leq 0.05$, Student's $t$-test). See also Supplementary Figs 2–5.

our data indicate that day 8 CD34 + VECAD + cells are enriched for HE and can be used to investigate EHT transition and blood development. We next tested lymphoid differentiation of our HE population to assess multi-lineage potential.

**HE cells display multi-lineage differentiation.** Sorted CD34 + cells were seeded on an OP9-Delta-Like 4 (OP9-DL4) stromal feeder layer, and cultured as described previously[28] (Fig. 2c). T-cell generation from hPSC-derived cells has previously been used to demonstrate the generation of haematopoietic progenitors during the second wave of embryonic haematopoiesis[29]. By day 15 of OP9-DL4 culture, CD34 + expression was downregulated and CD45 + cells, a subset of which expressed CD5 + CD7 + indicative of pro-T lymphoid cells[30], emerged (Fig. 2c). Further

T-cell maturation was demonstrated by the appearance of CD4 + CD7 + cells by day 25 (Fig. 2c). As a final characterization, cells from day 8 of our serum-free differentiation culture were sorted into HE-positive (CD34 + VECAD + CD43 − CD45 − ) and -negative (CD34 − VECAD − CD43 − CD45 − ) fractions, to test for short-term *in vivo* haematopoietic engraftment in mice. Positive and negative fractions from five differentiation cultures were pooled and frozen (Supplementary Fig. 4a). After thawing, purity analyses of pre-transplanted positive and negative fractions were conducted by flow cytometry (Supplementary Fig. 4b). These fractions were subsequently injected intravenously into sublethally irradiated mice. Ten weeks after transplantation, mice were killed and human-derived blood cells were identified using antibodies against human leukocyte and

CD45 antigens. Human CD45 + reconstitution in the peripheral blood was significantly higher in mice injected with the positive fraction (0.77%) compared to mice injected with the negative fraction (0.20%) ($P \leq 0.05$, $n = 3$, Student's $t$-test; Fig. 2d). Furthermore, frequencies of both CD41 + (megakaryocytes) and CD3 + (T cells) cells were highest in the peripheral blood of mice injected with the positive fraction compared to those injected with the negative fraction (summarized in Supplementary Fig. 4c; flow cytometry plots in Supplementary Fig. 5). These results demonstrate that day 8 hPSC-derived HE cells are capable of giving rise to T-cell-restricted lineage cells *in vitro*, and haematopoietic blood cells *in vivo* (albeit at low levels), indicative of their potential for definitive blood differentiation. We next set out to use this population of cells to explore conditions that modulate blood cell emergence.

**HE colony size and pitch impact blood progenitor yield.** Micropatterning of extracellular matrix (ECM) on cell culture surfaces enables the direct control of cell colony size and configuration. To control and study cellular organization during EHT, day 8 hPSC-derived HE cells were seeded on ultraviolet-treated ECM micropatterned 96-well plate surfaces, and CD45 + blood cell emergence was tracked over 5 days (schematic depicted in Fig. 3a). These micropatterned surfaces allowed us to control the local microenvironment by manipulating colony size, colony pitch (distance between colony centres) and colony clustering (the density for fixed number of colonies). Colony size was varied, independent of colony pitch, by increasing colony diameter at a fixed pitch (representative images in Figs 3b,c). Colony pitch was manipulated by maintaining colony diameter and varying distance between colony centres. Colony clustering was controlled by configuring colony size and spacing to keep the total number of cells within a well constant (constant global cell density). Representative images and average colonies per manipulation are summarized in Supplementary Fig. 6a and b, respectively. To ensure that patterning conditions were not selecting for CD34 + VECAD + cells, analysis was performed at 6 h and 5 days post seeding. Similar patterned cell densities were observed across all treatments before the onset of CD45 + blood generation (Supplementary Fig. 7). Intra-colony densities in increasing colony size and increasing pitch (Supplementary Fig. 7e) showed no significant changes throughout the assay ($t = 6$ h to 5 days post seeding).

To investigate the effect of colony size on blood induction, hPSC-derived HE (from day 8 hPSC aggregates) were seeded as colonies of 150 μm diameter and 500 μm pitch (hereafter referred as 150–500), 200–500 and 400–500 and cultured in haematopoietic medium to generate CD45 + blood cells (representative flow cytometry plots in Supplementary Fig. 7c). We found that blood cell induction decreased by $\geq 2$-fold from 150–500 compared to larger colony size conditions ($P \leq 0.05$, $n = 3$, one-way ANOVA with *post hoc* Tukey test; Fig. 3d). Interestingly, CD34 + VECAD + expression in HE colonies after 5 days of induction was significantly higher in 150–500 micropatterns compared to 400–500 micropatterns (Supplementary Fig. 7d). Moreover, the 150–500 colony size resulted in a 5.5-fold increase in CD45 + blood generation ($P \leq 0.05$, $n = 3$, Student's $t$-test; Supplementary Fig. 6c) and 2-fold increase in CFC frequency ($P \leq 0.05$, $n = 3$, Student's $t$-test; Supplementary Fig. 6c) compared to non-micropatterned conditions. These results indicate that smaller colonies exhibit enhanced CD45 + generation compared to larger colonies and non-micropatterned treatments.

To investigate a wide range of colony pitches, small colonies with 200 μm diameters were generated at varying pitch lengths of 400, 500 and 800 μm. CD45 + blood cell induction was enhanced

with increasing pitch with colonies of 200–800 generating 2.5-fold more CD45 + blood cells compared to 200–400 colonies ($P \leq 0.05$, $n = 3$, one-way ANOVA with *post hoc* Tukey test; Fig. 3d). CD34 + VECAD + expression did not appear to deviate with changes in colony pitch (Supplementary Fig. 7d). These results demonstrate that colonies of the same size but with greater pitch enhance CD45 + blood generation compared to colonies with smaller pitch. To test the effect of clustering on blood induction, colony size was varied while maintaining constant colony coverage (calculated at ∼12.5% of total area within well). CD45 + blood cell induction was similar between all treatments (Fig. 3d). These results indicate that at constant coverage, colony clustering (irrespective of colony size) does not enhance haematopoietic induction.

We next tested gene expression levels of haematopoietic-associated factors in blood cells generated from 150–500 and 400–500 micropatterned colonies. We assayed *TAL1*, *MYB*, *NR2F2*, *EPHRINB2*, *NOTCH1*, *KDR*, *DLL4* and *JAG1* expression via quantitative real time–PCR on day 0, 2 and 5 post patterning. It was found that *MYB* and *TAL1* gene expression was similar between blood cells generated from 150–500 and 400–500 colonies. Definitive haematopoietic cells have been postulated to arise primarily via blood vessel specification with the Notch and VEGF pathway being instrumental in this process. Day 5 blood cells generated from 150–500 micropatterns displayed higher expression of *NOTCH1* ($1.66 \times$), *EPHRINB2* ($3.53 \times$, $P \leq 0.05$, $n = 3$, Student's $t$-test), *KDR* ($2.65 \times$, $P \leq 0.05$, $n = 3$, Student's $t$-test), *DLL4* ($2.04 \times$), *NR2F2* ($3.3 \times$, $P \leq 0.05$, $n = 3$, Student's $t$-test) and *JAG1* ($3.5 \times$, $P \leq 0.05$, $n = 3$, Student's $t$-test) compared to 400–500 micropatterns (Fig. 3e). In addition to gene expression, we utilized the RUNX1C-GFP HES3 cell line to track haematopoietic differentiation. The RUNX1C isoform has been shown to be an important marker to identify definitive blood cells *in vivo*[31,32], and cells expressing this marker give rise to multipotent blood progenitor cells *in vitro*[33]. We patterned 150 and 400 μm diameter HE colonies in EHT cultures and assessed for CD45 + CD34 + RUNX1C + blood expression by flow cytometry. Our results indicate that blood cells generated from 150 μm colonies had seven fold higher CD45 + CD34 + RUNX1C + expression compared to blood cells from 400 μm colonies (representative flow cytometry plots are displayed in Supplementary Fig. 8). These results demonstrate that geometric restriction of HE cells affects blood cell progenitor yield and their respective gene expression patterns. Overall, our results demonstrate that blood cell generation is enhanced in smaller colonies that are farther apart, suggesting endogenous inhibitory factors influence haematopoietic differentiation from HE cells.

**Colony size controls IP-10 secretion.** Recent literature has indicated inflammatory molecules provide cues for the generation of blood progenitor stem cells[34]. To test whether the observed increase in CD45 + blood generation in lower-coverage micropatterns was the result of a reduction in inhibitory factors produced over 5 days, we used an inflammatory human cytokine magnetic bead panel to analyse secreted factor concentrations in the media of micropatterned and non-micropatterned cultures. This analysis revealed that IP-10 was significantly elevated ($4 \times$) in non-micropatterned treatments ($0.017 \pm 0.0070$ pg ml$^{-1}$ per cell) compared to micropatterned conditions ($0.0040 \pm 0.0012$ pg ml$^{-1}$ per cell; $P \leq 0.05$, $n = 3$, Student's $t$-test; Fig. 3f). We next tested whether varying IP-10 concentration in differentiating HE cells affects blood generation. Supplementation of exogenous IP-10 to both 150–500 and 400–500 μm treatments leads to significant reduction (65%) in blood generation compared to respective control treatments (Fig. 3g). To

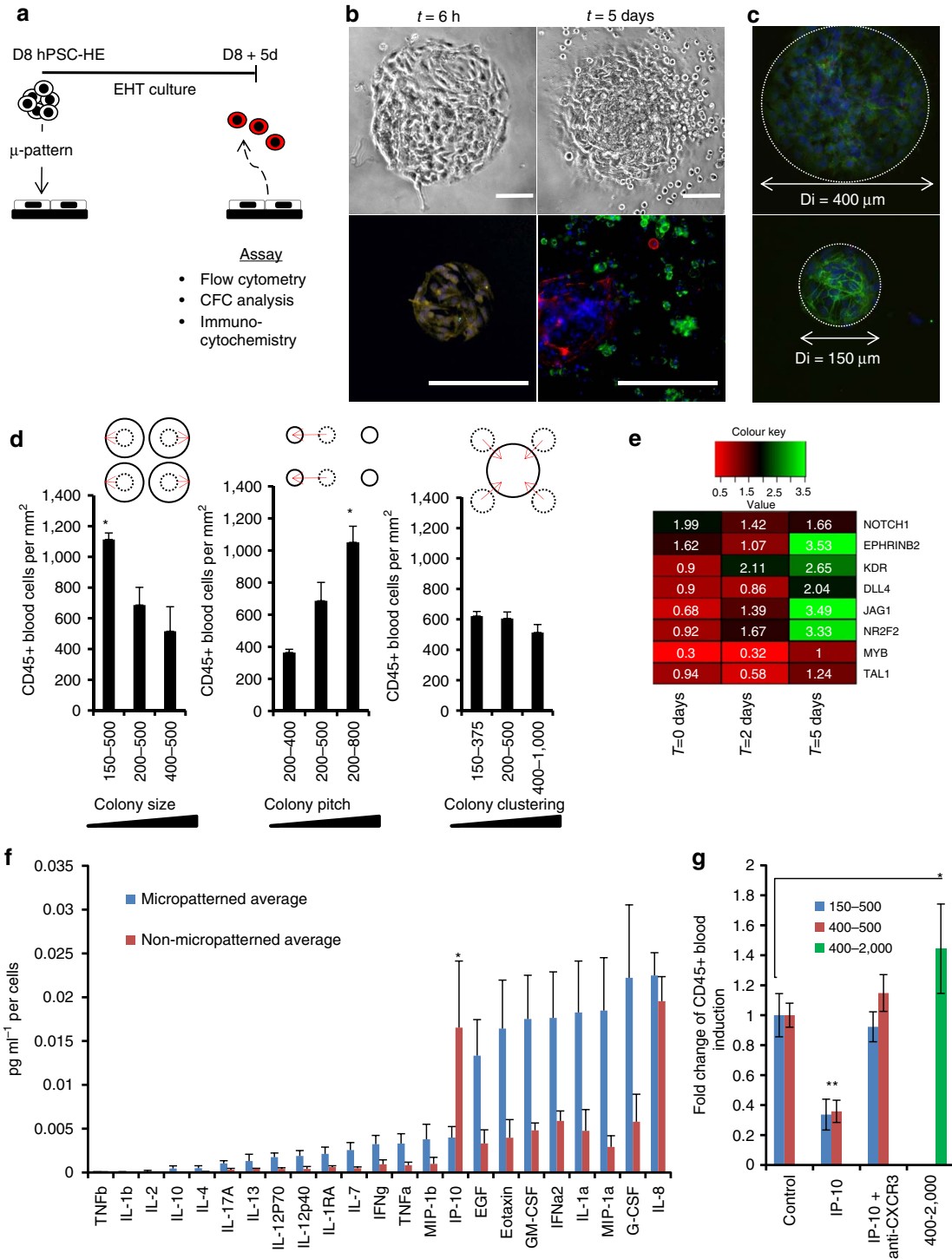

**Figure 3 | Spatial control of day 8 hPSC-derived HE via micropatterning enhances CD45+ generation in the 150–500 condition during haematopoietic differentiation.** (**a**) Schematic of EHT culture and assessment of cells at assay end point. Micropatterns are depicted as diameter–pitch (diameter: diameter of colony/pitch: distance between colony centres; for example, 150–500). (**b**) Micropatterned day 8 prospective HE colonies cultured in haematopoietic-inducing medium on 400–500 μm and 150–500 μm stamps at $t = 6$ h and 5 days post seeding (yellow—VECAD; red—CD34; green—CD45; blue—Hoechst). (**c**) Fluorescent images of 400 and 150 μm diameter patterned hPSC-derived prospective HE colonies (green—VECAD; blue—Hoechst). (**d**) CD45+ blood cell generation per cultured area among micropatterns in increasing colony size, pitch and clustering at $t = 5$ days, $n = 3$. (**e**) Gene expression of positive haematopoietic factors is enhanced in blood cells generated from 150–500 colonies compared to 400–500 colonies, $n = 3$ (~450 pooled colonies per time point). (**f**) IP-10 is an inhibitory chemokine produced during haematopoietic induction; conditioned media analysis of micropatterned and non-micropatterned treatments using an inflammatory panel of chemokines. IP-10 concentration was significantly enhanced in non-micropatterned treatments, $n = 3$. (**g**) IP-10 supplementation in 150–500 and 400–500 micropattern cultures decreases CD45+ blood induction from hPSC-derived HE cells, whereas supplementation with anti-CXCR3 antibody rescue haematopoietic deficiencies. Moreover, haematopoietic induction can be significantly enhanced by culturing cells as lower-coverage colonies, $n = 3$. Data are presented as mean ± s.e.m. Treatments displaying asterisks are significantly different from other groups ($P \leq 0.05$, Student's $t$-test or one-way ANOVA *post hoc* Tukey test). Scale bar, 200 μm. See also Supplementary Figs 6–10.

circumvent IP-10 inhibition, anti-CXCR3 (receptor to IP-10)-neutralizing antibodies were added to micropatterned differentiating HE cultures over 5 days (Fig. 3g). This treatment moderately increased haematopoietic induction in high-coverage 400–500 μm treatments compared to respective controls. Previously, IP-10 has been reported to signal through the p38 mitogen-activated protein kinase (p38 MAPK) pathway[35] and thus we explored activation of phosphorylated p38 in HE-optimized 150–500 conditions on day 5 across control, IP-10, anti-CXCR3 and VX-702 (p38 MAPK inhibitor) treatments (Supplementary Fig. 9a). We observed a moderate, but not significant, increase in VECAD + p-p38 + -expressing cells in the presence of IP-10 ($1.1 \pm 0.06$-fold change over control treatments; Supplementary Fig. 9b). However, anti-CXCR3 and VX-702 treatments significantly reduced p-p38-expressing cell numbers compared to control conditions ($0.69 \pm 0.07$ and $0.04 \pm 0.008$-fold change, respectively, $P \leq 0.05$, $n = 3$, one-way ANOVA with post hoc Tukey test; Supplementary Fig. 9b). Importantly, although IP-10 treatment moderately increased VECAD + p-p38 + cell levels, CD45 + cell induction was significantly reduced compared to control treatments (Supplementary Fig. 9b). When IP-10 was supplemented with anti-CXCR3 antibody and VX-702, CD45 + cell induction was rescued ($0.83 \pm 0.02$- and $1.13 \pm 0.09$-fold change to control treatments, respectively; Supplementary Fig. 9b). VX-702-treated conditions were also tested using CFC assays (Supplementary Fig. 9), demonstrating a similar rescue from IP-10 treatment as the CD45 + output. Our results indicate that in the presence of IP-10, CD45 + induction can be rescued by decreasing p-p38 levels. Interestingly, although anti-CXCR3 treatment reduced p38 phosphorylation, enhancement in CD45 + blood generation (compared to control) was not significantly enhanced, perhaps due to effects of other endogenous inhibitors. To investigate our hypothesis, we cultured HE cells as colonies with 400 μm diameter and 2,000 μm pitch (400–2,000) to further decrease overall cell density (theoretical well coverage = 3%) compared to 150–500 (theoretical well coverage = 7%) and 400–500 (theoretical well coverage = 50%). Our results show that CD45 + haematopoietic induction from 400–2,000 colonies was significantly better than 150–500 ($1.4 \times$; Fig. 3g) and 400–500 ($3.2 \times$; Fig. 3g) colonies inferring that endogenous inhibitors were further mitigated by culturing HE cells as lower-coverage micropatterns. Together these results identify IP-10 as a molecule that inhibits hPSC-derived blood induction and whose secreted concentration is colony size-controlled.

To further assess the biological relevance of IP-10 during human haematopoiesis, we cultured magnetically sorted CD34 + cells from umbilical cord blood cells for 7 days. We used flow cytometry to assess HSC (CD34 + CD45RA − CD38 − CD90 + and CD34 + CD45RA − CD38 − CD90 + CD49f + ), progenitor cell (CD34 + CD45RA + CD90 + , CD34 + CD45RA + CD90 − and CD34 + CD45RA − CD90 − ) and differentiated cell (CD34 − ) yields. Our results indicate that IP-10 supplementation to CD34 + cord blood (CB) cells significantly decreases the HSC phenotype ($P \leq 0.05$, $n = 3$, Student's t-test) compared to control treatments (Supplementary Fig. 10a). Similar to our hPSC study, addition of anti-CXCR3 antibody with IP-10 rescued HSC yields to levels comparable to control treatments. Moreover, since IP-10 signals through the p38 MAPK signalling pathway, we hypothesized that addition of VX-702 (p38 MAPK kinase inhibitor) could potentially enhance the HSC phenotype yield. The addition of VX-702, either alone or in conjunction with anti-CXCR3 antibody, rescued the HSC phenotype to levels that were comparable to control treatments. Together, our results extend our data to adult definitive haematopoiesis and demonstrate that IP-10 is a potent inhibitor of HSC-enriched cells in cord blood

and its effect can be mitigated by using anti-CXCR3 antibody and VX-702 treatment.

**Colony size affects HE organization and CD45 + emergence.** Our observation that endogenous signalling may impact EHT, along with recent observations from our group and others of the role of spatial gradients in cell fate patterning[11,36], led us to examine the role of HE spatial organization on the emergence of hPSC-derived human blood progenitors. Post-imaging analysis was used to interrogate spatial CD34 and VECAD expression in HE colonies at various colony sizes (schematic depicted in Fig. 4a). Pronounced radial organization of CD34 + expression was observed in larger colonies (200 and 400 μm diameter), wherein protein expression was higher in colony centres compared to peripheral cells (Fig. 4b). Moreover, radial organization was weaker as colony size decreased to 150 μm, consistent with smaller-sized colonies having smaller endogenous signalling gradients[11,13]. Importantly, no gradients were detected in colonies cultured in base medium devoid of haematopoietic cytokines (Fig. 4b) implying that radial organization only occurs during haematopoietic induction. To eliminate false-positive signals from variable intra-colony cell densities, we assessed for β-actin expression and found similar signals across 150–500 and 400–500 micropatterns on day 5 (Supplementary Fig. 7a). To quantify this intra-colony spatial organization, CD34 + and VECAD + intensity values were binned into six groups corresponding to distances from respective colony centres and then normalized to the bin closest to the centre. It was found that CD34 + and VECAD + expression levels were uniform throughout each colony in the 150–500 condition, whereas in the 200–500 and 400–500 conditions, protein expression levels were significantly greater in colony centres compared to colony edges ($P \leq 0.05$, $n = 3$, Student's t-test; Fig. 4c). These results indicate that manipulating HE colony size spatially regulates CD34 and VECAD expression.

To assess whether colony size impacts spatial control of haematopoietic cell emergence, we tracked blood cell-budding frequencies (emerging CD45 + cells) in inner and outer areas (inner: <50% colony radius; outer: >50% colony radius) of 150–500 and 400–500 colonies using live imaging microscopy (Supplementary Movies 1 and 2, representative image in Supplementary Fig. 7b). Blood cell-budding frequencies were 4.5-fold higher in colony centroids than in colony edges in 400–500 colonies, and comparable in colony centroids and edges in 150–500 μm colonies (Fig. 4d). These data indicate that geometric restriction of HE cells can spatially impact intra-colony marker expression and blood cell emergence.

## Discussion
Embryonic haematopoiesis is a dynamic process guided by temporal and spatial cues. The dorsal aorta is the first site of definitive blood cell emergence during embryogenesis. Interestingly, HE cells that give rise to these blood cells are spatially restricted to the ventral floor and are surrounded by stromal cells containing a mix of endothelial and mesenchymal cells[8]. Moreover, HE cells—situated in a region of densely packed cells[37], ECM proteins[37] and polarized cytokine gradients[38]—receive signals from surrounding tissues for intra-arterial blood cluster formation[9,39]. Mimicking individual niche components not only elucidates how blood emergence occurs but also provides information that can improve in vitro differentiation protocols to generate target cell types from hPSCs. Here we present a robust protocol for hPSC-derived HE cells using size-controlled aggregates and chemically defined medium, which consistently gives rise to myeloid and pro-T cells

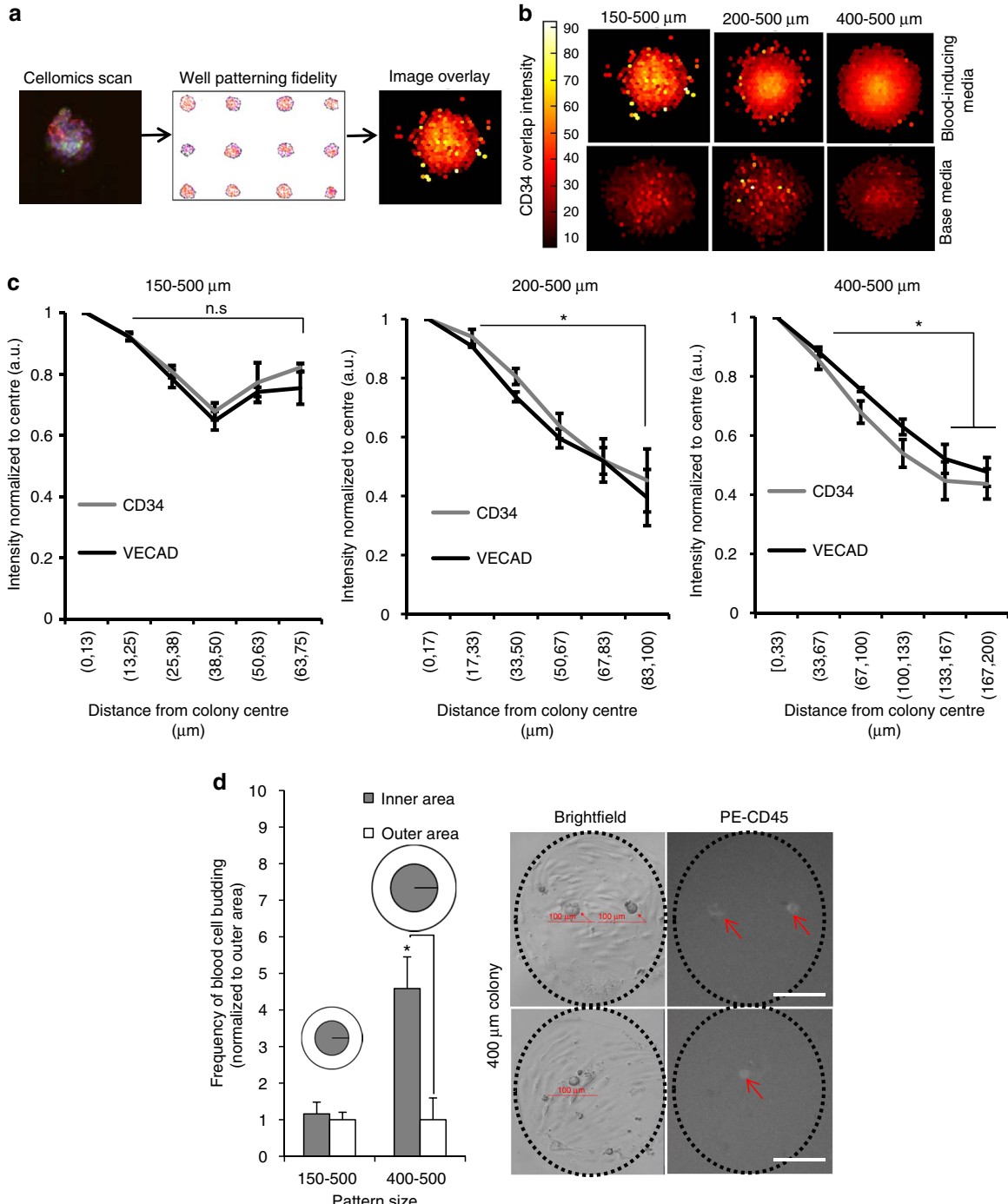

**Figure 4 | Spatial control of day 8 hPSC-derived HE via micropatterning influences intra-colony organization of CD34 and VECAD expression and blood cell emergence during haematopoietic differentiation.** (**a**) Overview for generating CD34+ and VECAD+ distribution overlay. Fluorescent images of colonies in unique treatments are assessed for patterning fidelity and then single-cell data are consolidated to generate superimposed images. (**b**) Representative CD34+ expression overlays in increasing colony size in haematopoietic-inducing or base medium (as outlined in Methods). (**c**) Quantification of CD34 and VECAD expression in increasing colony size from the centroid to the edge, $n = 3$. (**d**) Live imaging analysis comparing budded CD45+ blood cell frequencies between the inner and outer colony areas of 150 and 400 μm colonies (left), $n = 3$. Data are presented as mean ± s.e.m. Treatments joined by asterisks are significantly different ($P \leq 0.05$, Student's $t$-test). Scale bar, 100 μm. See also Supplementary Movies 1 and 2.

*in vitro* and display short-term reconstitution *in vivo*. Further, we provide evidence that micropatterning size-restricted hPSC-derived HE colonies significantly increases CD45+ blood generation, with higher blood cell emergence frequencies in smaller and more separated colonies. These configurations mitigate the inhibitory effects of endogenous molecules such as

IP-10 on blood development. Mechanistically, we link the p38 MAPK pathway to the decrease of CD45+ blood generation via IP-10 supplementation, and extend this data set to cord blood-derived cells. Finally, we show that gradients of blood cell emergence occur in spatially restricted HE colonies with larger diameters (>200 μm).

To develop an *in vitro* model of EHT, which would allow us to study the role of spatial organization and endogenous signalling during definitive blood development, we designed and optimized a protocol to derive HE cells from hPSC. In this protocol, we control hPSC aggregate size in serum-free differentiation conditions to efficiently generate HE cells. The optimized aggregate size for induction to definitive HE phenotypes (500 cells per aggregate) yielded 19-fold higher expression of the definitive HE phenotype compared to current protocols[25]. Our HE cells possessed multi-lineage potential consistently giving rise to myeloid and lymphoid lineages (CD5 + CD7 + pro-T cells), cardinal features of definitive haematopoiesis. Moreover, our candidate HE population (sorted CD34 + VECAD + CD43 − CD45 − ) was capable of short-term reconstitution in non-obese diabetic/SCID/IL-2Rgc-null (NSG) mice and harboured the majority of blood-inducing cells, as this fraction showed significantly higher engraftment than the negative fraction. Although low levels of mean CD45 + engraftment (0.77%) were observed, these levels fall within the upper range of previously reported engraftment levels by Tian *et al.* (0.16–1.44%)[40] and Wang *et al.* (<1%)[41]. Ledran *et al.*[42] have reported CD45 + engraftment levels close to 16%, however these hPSC-derived cells were cultured on AGM-derived stroma, thereby negating clinical use. Higher CD45 + reconstitution levels in primary recipients have been recently reported but in these cases hPSC-derived cells were genetically modified[43]. Low engraftment of hPSC-derived cells may be due to functional differences between embryonic blood progenitors and mature definitive HSCs. For example, the absence of CXCR4 expression on hPSC-derived cells is a requirement for proper homing to the haematopoietic organ. To circumvent this issue, studies have injected hPSC-derived cells intra-femorally into immunocompromised mice as the femur is a more conducive environment for haematopoietic reconstitution[43]. Interestingly, our studies showed undetectable reconstitution levels in bone marrow samples compared to levels observed in peripheral blood indicating that embryonic-like blood progenitors lack homing capabilities or may home to secondary haematopoietic organs in adult recipient mice. Although engraftment rates of the negative fraction are 3.5-fold lower compared to the positive fraction, the ability of the negative fraction to minimally reconstitute immunocompromised mice could be due to maturation to the CD34 + VECAD + phenotype at time of injection or due to *in vivo* microenvironments conferring haematopoietic fates on injected hPSC-derived cells as has been previously reported[44]. By controlling culture parameters such as aggregate size and oxygen tension, we present a scalable, serum-free differentiation platform that robustly generates hPSC-derived HE with multi-lineage potential.

Determining the signalling pathways, which are regulated by colony size and spacing, that impact blood cell emergence was of particular interest. We have previously shown that micropatterning influences autocrine and paracrine signalling during PSC maintenance and differentiation[11,13,14], and speculate that endogenous control over signalling pathways plays an important role in the haematopoietic differentiation system described here as well. Consistent with our data set, smaller islands of micropatterned hPSCs have been reported to be efficient in endothelial differentiation compared to larger islands[45]—perhaps due to lower levels of inhibitory factors. We observed that IP-10 endogenously inhibited hPSC-derived haematopoietic induction in non-micropatterned conditions, and its concentration can be attenuated by configuring HE cells into smaller colonies, rescuing haematopoietic differentiation. A similar effect was observed when we supplemented CD34 + sorted cord blood-derived cells with IP-10. It has been shown that IP-10 inhibits human

endothelial cell motility and tube formation[46], which could lead to improper patterning of HE intermediates thereby resulting in decreased blood cell formation. We observed that IP-10-mediated haematopoietic inhibition was controlled via the p38 MAPK signalling pathway, which has been reported to play a crucial role in promoting *ex vivo* expansion of human cord blood HSCs[47]. Our results are the first to identify the role of p38 MAPK signalling pathway in generating haematopoietic cells from PSC-derived sources, and motivate future studies on the role of this pathway in *de novo* blood development. Interestingly, IP-10 knockout mice are not known to exhibit haematopoietic defects[48]. To explore a potential role for IP-10 during murine embryonic haematopoiesis, we mined literature for IP-10 and CXCR3 gene expression in haemogenic cells during EHT in the AGM. RNA-sequencing analysis of single-cell and 10-cell sorted haemogenic populations during murine AGM haematopoiesis[49] revealed the highest IP-10 and CXCR3 expression was present in mature HSC subtypes (defined as T2 pre-HSCs; Supplementary Fig. 10b). Supporting this data set, microarray analysis from E 11.5 AGM tissues[34] and RNA-sequencing analysis from E 10.5 AGM tissues[50] indicate that CXCR3 expression is enhanced in the HSC compartment compared to endothelial subtypes. Our data support a hypothesis that IP-10 can inhibit haematopoietic development either autonomously or non-autonomously by interacting with CXCR3 receptors present on HSCs. Inflammatory signals such as interferon gamma (IFNγ) and tumour necrosis factor-α (TNF-α) are greatly affected in IP-10-deficient mice, drastically impacting lymphoid cell circulation[51]. As inflammatory signals are important for murine embryonic haematopoiesis, IP-10 may play a role in modulating these signals. For example, in IP-10 $^{-/-}$ mice, there was an increase in IFNγ-producing CD8 + T cells compared to wild-type mice[51]. These results suggest IP-10 can modulate IFNγ expression in a feedback mechanism. During embryonic haematopoiesis, IP-10 could attenuate pro-inflammatory signals since chronic exposure to IFNγ has been reported to be detrimental for HSC function[52]. In our studies, blood cells generated from smaller colonies displayed higher expression levels of definitive haematopoietic factors such as *KDR*, *NOTCH1* and *JAG1* compared to higher-coverage micropatterns. This is in line with previous reports, which have indicated that endogenous signalling can skew haematopoietic differentiation. Temporal VEGFA inhibition during mouse embryonic stem cell haematopoietic differentiation is crucial for promoting CFC generation at the expense of endothelial differentiation[53]. Inhibition of transforming growth factor-β and hedgehog signalling can enhance blood progenitor output, suggesting that negative feedback from differentiating cells plays a key role in haematopoiesis[54]. Studies in murine and zebrafish models have also revealed roles for endogenous factors such as bone morphogenetic factor-4 (ref. 38), IFNγ (ref. 34) and phenylephrine[39] in AGM haematopoiesis. By engineering the cellular environment using micropatterning technology, we can further investigate endogenous factors that are crucial for hPSC-derived HE differentiation to definitive blood cells.

Collectively, the findings presented here demonstrate that engineering the haematopoietic developmental niche can impact blood cell yields and reveal parameters and molecules that control blood cell emergence. Our system provides a promising platform to investigate signalling pathways that inhibit blood cell emergence from differentiating hPSCs. Additional niche parameters such as immobilized ligands[8,55] and synergistic cell types[56] that are important for embryonic haematopoiesis can also be integrated into our system to closely mimic *in vivo* niches. Notably, our current results and the techniques we have described here represent crucial steps in

unravelling the underlying mechanisms that drive the generation of hPSC-derived HE cells to HSCs.

## Methods

**Maintenance of hPSCs.** The hESC lines HES2, H1, H9 and RUNX1C-HES3 (provided by A. Elefanty and E. Stanley—Monash University) have been authenticated by karyotyping and tested for mycoplasma contamination. Cells were maintained on irradiated mouse embryonic fibroblasts and cultured for 7 days in maintenance medium comprising Dulbecco's minimum essential media (DMEM)/F12 (78.5% v/v, Invitrogen), Knockout Serum Replacement (19% v/v, KOSR, Invitrogen), penicillin/streptomycin (0.5% v/v, Invitrogen), non-essential amino acids (1% v/v, Invitrogen), β-mercaptoethanol (1% v/v, Invitrogen) and 10 ng ml$^{-1}$ basic fibroblast growth factor (bFGF; R&D). Cells were maintained at 37 °C humidified air with 5% $CO_2$ and 21% $O_2$ with daily medium exchange.

**Differentiation of hPSCs to HE.** Differentiation was initiated by forming hPSC/hiPSC aggregates of controlled cell numbers in microwell plates (Aggrewell 24 well, StemCell Technologies) manufactured in-house using 400 μm polydimethylsiloxane inserts cast from a silicone master mould and sterilized as previously described[15]. HPSC colonies were dissociated with 5 min TrypLE Express (Invitrogen) treatment and plated onto Geltrex- (diluted 1:50, Invitrogen) or Matrigel (diluted 1:30, Thermo-Fisher)-coated six-well plates at a split ratio of 1:3 for 48 h. Single-cell suspensions of mouse embryonic fibroblast-depleted hPSCs were generated by 5 min TrypLE Express treatment and resuspended in 50% FBS in DMEM/F12 followed by two washes in DMEM/F12. Cell counts were performed to calculate the volume of cell suspension required to generate specific aggregate sizes. The specified cell numbers were then resuspended in medium supplemented with ROCK inhibitor Y-27632 (RI) (1:1,000, Sigma-Aldrich) and transferred to microwell plates. The microwell plates were centrifuged at 1,500 r.p.m. for 5 min to force cell aggregation in individual microwells. The plate was then placed in a humidified incubator set at 37 °C, 5% $CO_2$ and 5% $O_2$ (hypoxia). Cell aggregates were cultured for 25 days (inclusive) and supplemented with cytokines in a two-stage approach in base medium comprising StemPro34 (Invitrogen), ascorbic acid (50 μg ml$^{-1}$; Sigma), L-glutamine (1% v/v, Invitrogen), penicillin/ streptomycin (1% v/v), 1-monothioglycerol ($4 \times 10^{-4}$ M; Sigma) and transferrin (150 μg ml$^{-1}$; Roche). For HE induction (10 days), cells were cultured in medium comprises BMP4 (40 ng ml$^{-1}$, R&D), vascular endothelial growth factor (VEGF; 50 ng ml$^{-1}$, R&D), stem cell factor (SCF; 40 ng ml$^{-1}$, R&D) and bFGF (5 ng/ ml$^{-1}$, Peprotech). For haematopoietic induction (10–25 days), cells were cultured in haematopoietic-inducing medium comprising SCF (50 ng ml$^{-1}$), VEGF (50 ng ml$^{-1}$), interleukin (IL)-3 (30 ng ml$^{-1}$, R&D), IL-6 (30 ng ml$^{-1}$, R&D), Epo (3 U ml$^{-1}$, R&D), Tpo (30 ng ml$^{-1}$, R&D) and bFGF (5 ng ml$^{-1}$, R&D). For specific studies, SB-431542 (6 μM) and CHIR99021 (3 μM) were added to the differentiation medium from day 2 to 3. Cultures were maintained in hypoxia (5% $O_2$) from day 1 to 10 (inclusive) and switched to normoxia from day 11 to 25 (inclusive). On day 5, cell aggregates were transferred to low-attachment six-well plates, and medium was exchanged every 3 days.

**Ultraviolet-lithography micropatterning of culture plates.** To control the size and density of hPSC-derived HE colonies, ECM was micropatterned at spatially controlled geometries. Glass slides (Schott, 110 mm × 74 mm glass slide) were activated in a plasma cleaner and then coated with poly L-lysine-grafted-poly-ethylene glycol (SUSOS, 0.1 mg ml$^{-1}$). The glass slide was incubated at 37 °C in a humidified environment for 24 h. To create micropatterned islands, a photomask with specific geometries was activated in a plasma cleaner. The photomask was sandwiched with polyethylene glycol-treated glass slides and placed in an ultra-violet ozone (UVO) chamber (Jetlight) and activated for 10 min to create patterned surfaces. After deep ultraviolet etching, the photomask–glass slide superstructure was gently washed with ddH$_2$O and dried gently with N$_2$ gas. To construct 96-well patterned plates, 96-well bottomless plates (VWR) were coated with Loctite epoxy adhesive and placed on the activated glass slide to create a sealed patterned plate. The following colony geometries were generated using this process (diameter (μm)–pitch (μm)): 150–500; 200–500; 400–500; 200–400; 200–500; 200–800; 150–375; 200–500; and 400–1,000). For immobilization of proteins, ddH$_2$O was dispensed into the 96-well micropatterned plate and incubated for 30 min. N-(3-dimethylaminopropyl)-N'-ethylcarbodiimide hydrochloride (15 mg ml$^{-1}$, Sigma) was dissolved in ddH$_2$O and aliquoted at 100 μl per well for 20 min. The wells were washed with ddH$_2$O twice. Next, 100 μl of an ECM mixture comprising fibronectin–gelatin (0.00125% fibronectin/0.002% gelatin) in ddH2O was added to each well and the plate was incubated for 3 h to allow patterned ECM islands to form. The ECM mixture used was previously published as an appropriate substrate for screening pluripotency regulators on micropatterned hPSCs[57,58]. After the incubation, the wells were washed twice with PBS and stored at room temperature until initiating cell culture.

**Endothelial to haematopoietic transition assay.** Day 8 hPSC-derived HE cells were dissociated as single cells using collagenase (Invitrogen) and TrypLE Express as described above. Cells were strained through a 40 μm cell strainer, centrifuged

and resuspended in StemPro-34 base medium with supplement and RI (1:1,000). Cells were seeded at $3.0 \times 10^4$ cells per micropatterned well and incubated at 37 °C for 6–12 h, gently washed twice with PBS to remove unattached cells, then incubated in haematopoietic inducing medium for 5 days to investigate blood generation from HE colonies. Haematopoietic induction from HE colonies was decreased by IP-10 supplementation (10 ng ml$^{-1}$, R&D). To mitigate endogenous IP-10 secretion from HE colonies, anti-CXCR3 (0.2 μg ml$^{-1}$, R&D) was added every 24 h for 5 days as previously described[35].

**Haematopoietic CFC assay.** To assess blood progenitor potential, hPSC-derived cells or CD45 + blood cells generated from micropatterns were seeded in duplicate at $1.0 \times 10^5$ or $1.5 \times 10^5$ cells per 35 mm Greiner dish containing Methocult H4435 enriched medium (StemCell Technologies). Samples were scored based on morphology 14 days after the cells were plated according to the manufacturer's protocol.

**Endothelium analysis.** Sorted day 8 hPSC-derived HE cells were seeded onto fibronectin–gelatin-coated 96-well plates and allowed to settle for 4 h. Samples were washed twice with PBS and then incubated with DiI-Ac-LDL (Biomedical Technologies, 1:20) and 4′,6-diamidino-2-phenylindole (DAPI, 1:1,000) for 4 h at 37 °C to measure DiI-Ac-LDL uptake. To assess vWF production, samples were fixed with 4% paraformaldehyde (10 min at 37 °C), and then permeabilized with room temperature 100% methanol for 3 min. Samples were stained with primary mouse anti-human vWF IgG (1:20, BD Pharmingen) followed with secondary donkey anti-mouse Alexa Fluor 647 (1:200, Invitrogen) and DAPI (1:100, Invitrogen). Cells were washed three times with PBS, and images were taken using a Zeiss fluorescent microscope.

**Flow cytometry and cell sorting.** To evaluate HE differentiation by flow cytometric analysis, hPSC-derived cells were dissociated by incubation in 0.1% collagenase for 2 h followed by TypLE Express for 8 min. Cells were mechanically dissociated and resuspended in 2% v/v FBS in Hank's buffered saline solution (HF). The following antibodies (obtained from BD Biosciences unless otherwise specified) were used at indicated dilutions: CD34-APC (1:100); CD144-PE (4:100); CD144-V450 (4:100); CD43-PE (3:100); CD45-PE-CY7 (4:100); CD4-PE (1:50) or CD34-PE (1:50); CD5-PE/Cy7 (1:200); CD7-AF700 (1:200); CD43-APC (1:200); CD45-APC/eF780 (1:200, eBiosciences); Ckit-APC (1:100); and PE-KDR (1:50). Control samples were prepared by incubating duplicate cell samples with respective fluorochome-labelled isotype antibodies. Samples were incubated with antibodies for 35 min on ice and then washed three times in cold HF. Viability discrimination was performed simultaneously using the viability stain 7-amino-actinomycin D at 1 μl ml$^{-1}$ (Molecular Probes). Samples were analysed on a Becton Dickinson FACS Fortessa machine using BD FACS Diva Software. Cells were sorted using a FACS Aria (BD Bioscience) cell sorter (Donnelly Centre for Cellular & Biomolecular Research).

**Immunocytochemistry and image validation.** Cells were fixed with 4% paraf-ormaldehyde (10 min at 37 °C) and permeabilized with 100% room-temperature methanol for 3 min. Cells were stained with the following primary human antibodies overnight at 4 °C: VE-Cadherin (1:80, Cayman Chemical); CD34 (1:80, BD Pharmingen); CD31 (1:80, BD Pharmingen); CD45 (1:80, BD Pharmingen); and phospho-p38 MAPK (Thr180/Tyr182, 1:100, Cell Signalling). Secondary staining was carried out using Alexa Fluor antibodies (1:200) incubated for 1 h at room temperature and protected from light. DAPI was used to visualize DNA content. Cells were imaged using the Cellomics Arrayscan VTI platform as previously described[57]. Colony analysis, pattern fidelity and spatial organization of cells was performed using in-house clustering algorithms (content explorer) as previously described[57].

**Analysis of NSG mouse engraftment.** For engraftment studies, 8-week-old female mice, housed in a barrier facility under stringent pathogen-free conditions in filter-top cages, were used. Animals were handled in sterile cross flow hoods and given access to sterile food and water. All animal studies were performed according to the approved University Health Network research ethics board protocols. Female NSG mice were sublethally irradiated (250 rad) 24 h before transplantation. Day 8 hPSC-derived HE cells were magnetically sorted for CD34 + (HE-positive) or FACS-sorted for CD34 + VECAD + CD43 − CD45 − (HE-positive) and CD34 − VECAD − CD43 − CD45 − (HE-negative) fractions and frozen down in cryopreservation vials until day of transplantation. Transplantation was carried out by injecting $2.3 \times 10^5$ pooled cells of each fraction into irradiated NSG mice via tail vein based on previously published literature[41–43]. On a random basis, a total of three mice were injected per fraction. Mice were killed 10 weeks post transplantation. Peripheral blood, bone marrow and spleen from each mouse were collected and prepared for flow cytometry analysis. Red blood cells were lysed with 0.8% ammonium chloride solution, and the remaining cells were washed in PBS containing 5% FBS. The following markers were evaluated using antibodies purchased from BD Biosciences unless otherwise stated: CD45-PE (1:200); CD45-APC (1:200, Beckman Coulter); HLA-ABC-FITC (1:200, Beckman Coulter); CD133-PE (1:200, Miltenyi Biotec); CD19-FITC (1:200); CD11b-PECy7 (1:200);

CD235a-FITC (1:100); CD3-APC (1:200); CD34-FITC (1:100); CD45RA-APC (1:200); CD41-PE (1:200); and CD71-FITC (1:200); 7-AAD was used to assess cell viability. Mice were scored positive for human repopulation if at least 0.1% of cells were positive for both human CD45 and human leukocyte antigen.

**OP9-DL4 co-culture for T-cell lineage differentiation.** CD34+ enriched progenitors were cultured on OP9-DL4 cells[59] at a density of $1.0–3.5 \times 10^5$ cells per well of a six-well plate, and grown in alpha-MEM supplemented with 20% FBS (Gibco), 50 µg ml$^{-1}$ phospho-ascorbic acid (Sigma-Aldrich), 5 ng ml$^{-1}$ rhIL-7, 5 ng ml$^{-1}$ rhFLT3L and 10 ng ml$^{-1}$ rhSCF (Miltenyi Biotec). Cells were fed every 2–3 days, and transferred to fresh OP9-DL4 stroma every 5 days.

**Live imaging and analysis.** To visualize and track emerging blood cells, day 8 hPSC-derived cells were dissociated as single cells, seeded as micropatterned colonies in 96-well plates and allowed to adhere for 6–12 h. Non-adherent cells were washed away and placed in 200 µl haematopoietic-inducing medium with PE-CD45 (1:400) for 5 days. Cells were cultured in a humidified 5% (v/v) $CO_2$ air environment at 37 °C housed in an on-stage incubation system with an inverted microscope (Zeiss). Phase-contrast images (acquired every 20 min) and fluorescent images (acquired every 2 h) were taken of representative colonies at ×10 objective using Zeiss AxioVision software. The X-cite 120 LED lamp (Lumen dynamics) was used for fluorescence illumination at 60% intensity. Images were analysed and manually tracked using ZEN blue software (Zeiss). For 400–500 µm patterns, quarter or half images were used as samples for calculating budding frequencies. Only cells with clear identity and behaviour were used for analysis.

**Quantitative real time–PCR.** Blood cells were generated from respective micropatterned colony treatments and pooled from three independent experiments (~450 colonies per time point). Total RNA was isolated and purified using PureLink RNA minikit (Invitrogen) according to the manufacturer's instructions. Normalized RNA across treatments were used to generated cDNA using Superscript III reverse transcriptase (Invitrogen) according to the manufacturer's instructions. Generated cDNA was mixed with primers and SYBR green mix (Roche) and run on the Applied Biosystems Quantostudio 6 Flex (Life Technologies). Relative expressions of genes were calculated using the delta–delta cycle threshold ($C_T$) method with the expression of GAPDH as an internal reference. Primer sequences are listed in Supplementary Table 1.

**Human conditioned media analysis.** Secreted levels of chemokines/cytokines were analysed in duplicate from conditioned media samples using an inflammatory panel against the human cytokine/chemokine 29-plex magnetic bead panel (Millipore, Billerica, MA, USA) with a Luminex 200 system (Luminex Co., Austin, TX, USA). Conditioned media samples were stored at −80 °C until use.

**Umbilical cord blood cell culture.** CD34+ cells were sorted using the EasySep CD34+ -positive selection enrichment kit (StemCell Technologies, Inc., Vancouver, BC, Canada). Briefly, CD34+ sorted cells were seeded at 1,000 cells per 96 U-bottom well in serum-free IMDM media (GIBCO, Rockville, MD) with 20% BIT serum substitute (StemCell Technologies) and 1% Glutamax (GIBCO). The media was supplemented with 100 ng ml$^{-1}$ SCF, 100 ng ml$^{-1}$ FMS-like tyrosine kinase 3 ligand (FL, R&D Systems), 50 ng ml$^{-1}$ thrombopoietin (R&D Systems) and 1 mg ml$^{-1}$ LDL (Calbiochem, La Jolla, CA). IP-10 was supplemented at 0.1 ng ml$^{-1}$. After 7 days of culture, cells were assessed for HSC phenotype (CD34+CD38−CD45RA−CD90+CD49f+), progenitor phenotype (CD34+ CD90+CD45RA+/CD34+CD90−CD45RA+/CD34+CD90−CD45RA−) and differentiated cell phenotype (CD34−).

**Statistical analysis and data representation.** Statistics were performed using OriginPro to identify significant trends between/among groups. A minimum sample replicate size of $n = 3$ was used for all data analysis, unless otherwise stated, for appropriate statistical testing. Comparisons were tested for normality before performing a one-way ANOVA with Tukey's *post hoc* analysis or Student's *t*-test. Treatments were significant using an alpha level of 0.05 with at least three independent replicates. Data are reported as mean ± s.e.m.

**Data availability.** The authors declare that the data supporting all the findings of this study are available within this paper and its Supplementary Information Files.

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

## Acknowledgements

We thank Dr Weijia Wang for running human conditioned media using the Luminex Co. system. We thank Dr Céline Bauwens for critical reading of this manuscript. We also thank Dr Andrew Elefanty and Dr Ed Stanley for providing us with the RUNX1C--GFP hESC cell line. We acknowledge the following funding sources: Canadian Institutes of Health Research and Medicine by Design; a Canada First Research Excellence Program at the University of Toronto (to P.W.Z. and J.C.Z.-P.); and Ontario Research Fund (to P.W.Z.). P.W.Z. is the Canada Research Chair in Stem Cell Bioengineering. J.C.Z.-P. is the Canada Research Chair in Developmental Immunology. N.R. received funding from Canadian Institutes of Health Research Doctoral Award and Ontario Graduate Scholarship in Science and Technology. P.M.B. was supported by a Canadian Institutes of Health Research (CIHR) Postdoctoral Fellowship award.

## Author contributions

N.R., P.M.B., T.U., J.C.Z.-P. and P.W.Z. designed research; N.R., P.M.B., L.H., M.T. and T.U. performed research; J.C.Z.-P. and P.W.Z. contributed new reagents/analytical tools; N.R., P.M.B., J.C.Z.-P. and P.W.Z. analysed data; and N.R. and P.W.Z. wrote the paper.

## Additional information

**Competing interests:** The authors declare no competing financial interests.

