## [Peer Review File · Nature Communications]

Reviewers' comments:

Reviewer #1 (Remarks to the Author):

The article entitled "Engineering the hemogenic niche mitigates endogenous inhibitory signals and enhances pluripotent stem cell-derived definitive blood progenitor generation" describes serum-free differentiation conditions to generate hemogenic endothelial (HE) cells from pluripotent stem cell (PSC) sources, interestingly by manipulation of colony size and spatial orientation to promote CD45⁺ cells. The HE cells were transplantable and engrafted into NSG mice. This system enabled the identification of interferon gamma induced 10 (IP-10) among a panel of cytokines as an inhibitor of CD45⁺ blood induction in the cultures. The investigation of an engineered system for efficient hemogenic induction would be of interest to the readers of Nature Communications and merits publication after the following points are addressed:

Major Points

1. The figure legend and methods section should describe the qRT-PCR data in Figure 3D more clearly. It seems that a similar number of colonies were analyzed to determine the ratio, but the exact method is unclear. Was the gene expression first normalized to housekeeping genes? Was the same cell number input used? In addition to the arterial genes provided (DLL4, EFNB2), it would be useful to include qRT-PCR expression analysis of hematopoietic genes MYB, TAL1 and RUNX1, and a venous gene such as NR2F2 in the different colonies.
2. How was the claim that anti-CXCR3 treatment reduced p38 phosphorylation quantified? Figure S8 shows representative images, but it is unclear how many colonies were scored and what specific metric was used to quantitate p38 levels. It is also hard to assess differences in levels between the control and anti-CXCR3 treated. More detail of how the levels were assessed is required. A Western blot showing p38 phosphorylation protein levels among the different conditions would also be helpful.
3. Regarding Figure 3F, the authors claim that other endogenous inhibitors may account for the lack of enhanced effect relative to controls. However, it is unclear whether anti-CXCR3 treatment alone is effectively blocking the receptor. Does combining IP-10 and anti-CXCR3 treatment lead to a change in blood induction relative to control, or would the anti-CXCR3 rescue IP-10 inhibition? How are p38 levels affected if cells were treated with both IP-10 and anti-CXCR3 by fluorescence or Western blot analysis?
4. Is IP-10 and its receptor expressed in vivo in the mouse AGM by immunohistochemistry? Even though the experiments were performed with human cells, it would be important to look at the mouse to confirm that IP-10 is involved during hematopoiesis to support its biological relevance.
5. Transplanted HE cells show relatively low engraftment based on peripheral blood sampling at 10 weeks. The low level of chimerism in the mice should be acknowledged in

the text with more discussion about it: the concern is that the in vitro conditions with this system are better than those obtained in vivo, and if the chimerism is not much better than what previous studies obtained using standard cultures, then the utility of this particular culture system becomes questionable.

Minor Points

6. The manuscript needs an English language edit in parts of the results and discussion sections (such as lines 204, 291, 298, 374, 443). Please confirm whether a 0.4 μm cell strainer was used (line 402) or correct the number as appropriate. The acronym mESC (line 77) is also not defined.
7. Please label the colors in Figure 2B in the images or figure legend.
8. A reference seems to be lacking regarding IP-10 signaling through p38 MAPK in line 235 (even though some relevant references are found later in the discussion).
9. The observation that IP-10 serves inhibitory roles is intriguing given that it is secreted in response to interferon- γ (IFN- γ). However, studies have suggested that inflammatory pathways mediated by IFN- γ are important for hematopoietic stem cell emergence in the AGM (work from the Daley, Speck, and Stainier labs). This warrants a brief discussion in text of the proposed inhibitory mechanism of IP-10 and how treatment with IP-10 would be biologically relevant if IFN- γ has an enhancing effect.
10. Please provide the total number of events captured or live cells analyzed in the FACS plots in Figure S5 (either in the figure legend or on the plots). I appreciate that the plots for all three recipients were provided.
11. Regarding the transplanted HE cells, were the cells frozen and thawed because limited numbers of cells were collected and required multiple rounds for pooling? The freezing process may affect the function of the cells in vivo. Do the transplanted cells persist long-term at 16 weeks and what is the associated long-term chimerism? Is engraftment also present in the bone marrow of recipient mice? That would confirm proper homing mechanisms.

Reviewer #2 (Remarks to the Author):

In this study Rahman et al. employ cell culture patterning technology to investigate how colony size and density affect the conversion of hemogenic endothelium to hematopoietic progenitors during differentiation of human pluripotent stem cells (hPSC). The main conclusions are that colony size and proximity can affect the frequency of generation of hematopoietic progenitors, implicating the microenvironment (and specifically CXCL10 or IP-10) in control of the specification and differentiation process.

The use of colony micropatterning to regulate endogenous signalling pathways crucial for

stem and progenitor cell self-renewal or differentiation is a strong approach to understanding the role of intercellular interactions in these processes. The study shows that progenitor cell yield can be influenced by colony size and patterning, and that these parameters likely impact on the production of cytokines and other extrinsic regulators of hematopoiesis. The culture system does not at this stage provide more than an incremental advance towards the generation of bona fide definitive hematopoietic cells but does nicely describe an alternate platform for mimicking an appropriate environment for the process. The experiments are carefully conducted and clearly presented. There are a number of specific points that the authors need to address to improve clarity and precision, as listed below.

Specific points:

1. P4 l 99 and Fig. 1a-do the authors mean low attachment rather than low cluster-I am not sure what a low cluster plate is
2. P5 l122-explain for a general readership the significance of CD73+ CD184+. Did the authors look at CD235 (Sturgeon et al. Nat. Biotech. 32 554,2010)?
3. P5 l132-VWF staining in Fig2b is feeble relative to the positive control, what does isotype control look like?
4. P6 l146-T cell generation is taken as an indicator of definitive hematopoiesis, but there is evidence (Boiers et al Cell Stem Cell 13: 535) that lymphoid cells emerge in the yolk sac prior to definitive hematopoiesis during development
5. P6 l158-sentence structure "purity analysis" was not "then injected"
6. P7 l64- CD41+ percentages do not appear to differ significantly between the two groups in S4C.
7. P7 l172- what was the composition of the ECM used, and what was the basis for the choice.
8. P7 l177-182-it might be preferable to refer to data showing that cell attachment was not affected by colony patterning here rather than later on
9. P8 l211-the modest differences in transcript levels for these factors would be more convincing evidence of the importance of local paracrine interaction if the authors had provided any information on protein levels. The significance of these data is not really very clear.
10. P8 l213 see comment 7 above
11. P9 l232-perhaps I have overlooked something but the effects of the AntiCXCR3 antibody are not very impressive, and it is unclear precisely how the data on 400-2000 colonies relate to this
12. P13 l310 and elsewhere-the authors should be careful about how they define definitive potential. This is a vexed area of PSC differentiation and claims must be carefully worded and justified.
13. P13 l317-the CXCL10 knockout apparently does not display hematopoietic defects; is it possible that this represents a difference in gene function between mouse and human?

Reviewers' comments:

Reviewer #1 (Remarks to the Author):

The article entitled "Engineering the hemogenic niche mitigates endogenous inhibitory signals and enhances pluripotent stem cell-derived definitive blood progenitor generation" describes serum-free differentiation conditions to generate hemogenic endothelial (HE) cells from pluripotent stem cell (PSC) sources, interestingly by manipulation of colony size and spatial orientation to promote CD45+ cells. The HE cells were transplantable and engrafted into NSG mice. This system enabled the identification of interferon gamma induced 10 (IP-10) among a panel of cytokines as an inhibitor of CD45+ blood induction in the cultures. *The investigation of an engineered system for efficient hemogenic induction would be of interest to the readers of Nature Communications and merits publication after the following points are addressed:*

Major Points

1. The figure legend and methods section should describe the qRT-PCR data in Figure 3D more clearly. It seems that a similar number of colonies were analyzed to determine the ratio, but the exact method is unclear. Was the gene expression first normalized to housekeeping genes? Was the same cell number input used? In addition to the arterial genes provided (DLL4, EFNB2), it would be useful to include qRT-PCR expression analysis of hematopoietic genes MYB, TAL1 and RUNX1, and a venous gene such as NR2F2 in the different colonies.

We apologize for the lack of details outlining the qRT-PCR data in our studies. To elaborate, blood cells were generated from respective micropatterned colony treatments and pooled from three independent experiments (~450 colonies per timepoint). Total RNA was isolated and purified using PureLink RNA minikit (Invitrogen) according to manufacturer's instructions. Normalized RNA across treatments were used to generate cDNA using Superscript III reverse transcriptase (Invitrogen) according to manufacturer's instructions. Generated cDNA was mixed with primers and SYBR green mix (Roche) and run on the Applied Biosystems Quantostudio 6 Flex (Life Technologies). Relative expressions of genes were calculated using the delta-delta cycle threshold (C_T) method with the expression of GAPDH as an internal reference. Primer sequences are listed in Table S1. We have included this detailed version in the methodology section of the manuscript.

As per the reviewer's suggestion, we have included qRT-PCR expression analysis of NR2F2 (venous gene) and hematopoietic genes (MYB and TAL1) between 150-500 and 400-500 micropatterned colonies. NR2F2 expression was 3.3-fold higher in 150-500 colonies compared to 400-500 colonies ($p \leq 0.05$, Figure 1A). In conjunction with EPHRINB2 expression, the results indicate blood cells generated from 150-500 micropatterned colonies arise from progenitors displaying robust blood vessel formation – a precursor for efficient endothelial to hematopoietic transition (EHT). In addition, it was found that MYB and TAL1 displayed similar gene expression in 150-500 colonies compared to 400-500 colonies by day 5 [Figure 1A].

To assess RUNX1C expression, we utilized a RUNX1C-GFP HES3 cell line (gifted by Andrew Elefanty and Ed Stanley, Monash University, Australia). The RUNX1C isoform has been shown to be an important marker to identify definitive blood cells in vivo (Swiers et al. 2010; Sroczyńska et al. 2009) and cells expressing this marker give rise to multipotent blood progenitor cells in vitro (Ditadi et al. 2015). We patterned 150 μm and 400 μm diameter hemogenic endothelium colonies in EHT cultures and assessed

for CD45+CD34+RUNX1C+ blood expression. Our results indicate that blood cells generated from 150 μm colonies had 7-fold higher CD45+CD34+RUNX1C+ expression compared to blood cells from 400 μm colonies [Figure 1B]. These results indicate that blood cells generated from 150 μm colonies are more representative of definitive progenitors found during embryonic hematopoiesis compared to cells from higher coverage hemogenic endothelium patterns. We have included this discussion in the results section.

Figure 1: Micropatterned colony size and density influence hematopoietic induction in differentiating PSCs. A) Gene expression levels of hematopoietic factors is enhanced in blood cells generated from 150-500 colonies compared to 400-500 colonies, $n = 3$ (~450 pooled colonies per timepoint). Bi) Fold change expression of CD45+CD34+RUNX1C+ of blood cells from 150-500 micropatterned colonies compared to 400-500 micropatterned colonies. Treatments that share asterisks

are significantly different ($p \leq 0.05$) Bii) Representative flow cytometry plots. Statistical significance was computed via Student's t-test. All error bars represent standard deviation. $N = 3$ biological replicates.

2. How was the claim that anti-CXCR3 treatment reduced p38 phosphorylation quantified? Figure S8 shows representative images, but it is unclear how many colonies were scored and what specific metric was used to quantitate p38 levels. It is also hard to assess differences in levels between the control and anti-CXCR3 treated. More detail of how the levels were assessed is required. A Western blot showing p38 phosphorylation protein levels among the different conditions would also be helpful.

We based our claim that anti-CXCR3 treatment reduced p38 phosphorylation on high throughput analysis of phospho-p38 (p-p38) levels using the Cellomics ArrayScan VTI platform [Figure 2A]. Colony analysis and pattern fidelity was performed using in-house clustering algorithms as previously described (Nazareth et al. 2013). Briefly, two biological replicates were performed and for each replicate data - 30 pooled colonies were analyzed. We observed a moderate, but not significant, increase in VECAD+p-p38+ expressing cells in the presence of IP-10 ($1.1 \pm 0.06\%$ fold over control treatments). However, anti-CXCR3 and VX-702 (a p38 MAPK inhibitor) treatments significantly reduced p-p38 expressing cell numbers compared to control conditions (0.69 ± 0.07 and 0.04 ± 0.008 respectively, $p \leq 0.05$, Figure 2Bi). Importantly, although IP-10 treatment moderately increased VECAD+p-p38+ cell levels, CD45+ cell induction was significantly reduced compared to control treatments. When IP-10 was supplemented with anti-CXCR3 antibody and VX-702, CD45+ cell induction was rescued (0.83 ± 0.02 and 1.13 ± 0.09 fold relative to control treatments respectively, Figure 2Bii). VX-702 treated conditions were also tested using colony forming cell (CFC) assays [Figure 2Biii], demonstrating a similar rescue from IP-10 treatment as the CD45+ output. Our results indicate in the presence of IP-10, CD45+ induction can be rescued by decreasing p-p38 levels. We have included our findings/discussion in the results section and the figure as Supplementary Figure 9.

Figure 2: Interferon gamma induced protein 10 (IP-10) reduction of CD45+ hPSC-derived blood progenitor cells can be rescued by inhibiting the p38 MAPK pathway. A) Representative fluorescent image of 150 μm diameter patterned hPSC-derived colonies after 5 days of hematopoietic induction among control (Ctrl), Ctrl + IP-10, Ctrl + IP-10 + anti-CXCR3 antibody, Ctrl + IP-10 + VX-702 (p38 MAPK inhibitor) treatments (green – VECAD; red – p-p38; blue – DAPI). Scalebar = 100 μm . B) Fold change in induction normalized to control of (i) VECAD+p-p38+ expressing cells, N = 2 biological experiments (ii) CD45+ expressing cells, N \geq 3 biological experiments and (iii) CFC generation, N \geq 3 biological experiments among stated treatments. Treatments that share asterisks are significantly different ($p \leq 0.05$). Statistical significance was computed using one-way ANOVA with post-hoc Tukey test (Bi) or Student's t-test (Bii and Biii). All error bars represent standard error mean. Scalebar = 150 μm .

3. Regarding Figure 3F, the authors claim that other endogenous inhibitors may account for the lack of enhanced effect relative to controls. However, it is unclear whether anti-CXCR3 treatment alone is effectively blocking the receptor. Does combining IP-10 and anti-CXCR3 treatment lead to a change in blood induction relative to control or would the anti-CXCR3 rescue IP-10 inhibition? How are p38 levels affected if cells were treated with both IP-10 and anti-CXCR3 by fluorescence or Western blot analysis?

To address the reviewer's query, we have cultured micropatterned hemogenic endothelium (HE) in the presence of IP-10 and IP-10 with anti-CXCR3 antibody. Comparing CD45+ induction in these two treatments to control conditions, our results indicate IP-10 supplementation significantly decreases CD45+ blood induction. Addition of anti-CXCR3 antibody to IP-10 cultures resulted in CD45+ induction comparable to control treatments [Figure 2Bii], as anticipated. Moreover, the p-p38 level was significantly reduced in the IP-10 supplemented condition compared to the IP-10 + anti-CXCR antibody condition [Figure 2Bi]. Xia et al (2016) have also reported the use of anti-CXCR3 antibody to block the effect of IP-

10 in their wound healing model. Our results demonstrate that anti-CXCR3 antibody supplementation can rescue IP-10 inhibition leading to CD45+ blood induction comparable to control treatments. Additional data relevant to this question can be found in Figure 2, for reference.

4. Is IP-10 and its receptor expressed in vivo in the mouse AGM by immunohistochemistry? Even though the experiments were performed with human cells, it would be important to look at the mouse to confirm that IP-10 is involved during hematopoiesis to support its biological relevance.

Our work supports a role for IP-10 in human cells. To assess the biological relevance of IP-10 during hematopoiesis, we cultured magnetically sorted CD34+ cells from umbilical cord blood for 7 days. We used flow cytometry to assess HSC (CD34+CD45RA-CD38-CD90+ and CD34+CD45RA-CD38-CD90+CD49f+), progenitor cell (CD34+CD45RA+CD90+, CD34+CD45RA+CD90- and CD34+CD45RA-CD90-) and differentiated cell (CD34-) yields. Our results indicate that IP-10 supplementation to CD34+ CB cells significantly decreases the HSC phenotype ($p \leq 0.05$) compared to control treatments [Figure Ai]. Similar to our hPSC study, addition of anti-CXCR3 antibody with IP-10 rescued HSC yields that were comparable to control treatments. Moreover, since IP-10 signals through the p38 MAPK signalling pathway, we hypothesized that addition of VX-702 (p38 MAPK kinase inhibitor) could potentially enhance HSC phenotype yield. Addition of VX-702, either alone or in conjunction with anti-CXCR3 antibody, rescued the HSC phenotype to levels that were comparable to control treatments. Although no enhancement in HSC yields were observed when VX-702 and anti-CXCR3 antibody were added; progenitor and differentiation yields were significantly lower compared to control and IP-10 supplemented treatments ($p \leq 0.05$) [Figure Aii and Aiii] indicating a conducive environment for HSC induction. These results extend our data to “adult” definitive hematopoiesis and demonstrate that IP-10 is a potent inhibitor of the HSC-enriched cells in cord blood and its effect can be mitigated by using anti-CXCR3 antibody and VX-702 treatment.

To further support the notion that IP-10 acts as an endogenous inhibitor for HSC induction, we cultured CD34+ sorted cord blood for 12 days in two different feeding regimes and assessed the concentration of secreted IP-10, total number of cells (TNC), total number of colony forming cells (CFC) and long term culture initiating cells (LTC-IC). In the first feeding regime (D=0, dilution of culture media via flow rate is nil) the culture volume is kept constant whereas in the second feeding regime (EXP) the culture volume increases exponentially. Our results show that the IP-10 concentration progressively increases over 12 days in D=0 feeding regimes [15-fold on day 12 compared to respective day 2, Figure 3Bi]. To counteract rising levels of IP-10, we diluted the medium over the course of 12 days (EXP feeding regime) and found that the IP-10 concentration remains relatively stable, only increasing 2.5 times from day 2 to day 12. The EXP regime leads to significantly higher progenitor cell yields as evidenced by greater CFC and LTC-IC output compared to the D=0 feeding regime [Figure 2Bii].

We have not investigated CXCL10 (IP-10)/CXCR3 protein expression during AGM hematopoiesis in mouse embryos because we believe it to be outside the scope of this study and do not make claims about the role of IP10 in non-human systems. Moreover, IP-10 knockout mice do not possess any hematopoietic defects as these mice only display apparent modulation in T-cell recruitment, trafficking and proliferation. As an alternative, we mined the literature for gene expression analysis of these markers in isolated hemogenic cells during the endothelial to hematopoietic transition in the aorta-gonad-mesonephros (AGM). To date, the highest resolution of gene expression data pertaining to prospective hemogenic cells during AGM hematopoiesis was generated by Zhou et al., 2016, Nature. We analysed the dataset in this publication which contained RNA-seq analysis of 10-cell and single cell blood progenitor populations. The 10-cell RNA-seq dataset from Zhou et al., 2016 revealed that the highest CXCR3 and CXCL10 (IP-10) expression was present in the most mature HSC subtypes (defined as T2

pre-HSCs) [Figure 2Ci]. Delving into the single cell RNA-seq data, it was evident that CXCR3 expression was also present in late T1 pre-HSCs and T2-pre HSCs – mature isolated HSCs – at E 10.5 [Figure 2Cii]. CXCL10 expression followed similar trends to CXCR3 expression in both 10-cell and single cell RNA-seq datasets. Collectively, these datasets reveal that CXCL10 and CXCR3 expression may play an important role in definitive embryonic hematopoiesis. Supporting this dataset, microarray analysis from E 11.5 AGM tissues (Li et al, Genes and Development, 2014) and RNA-seq analysis from E 10.5 AGM tissues (Kartalaei et al., JEM 2014) indicate that CXCR3 expression tends to localize in the HSC compartment compared to endothelial subtypes suggesting it has a specific role in modulating HSC maturation/proliferation [Analysis not shown]. Our data supports a hypothesis that CXCL10 can inhibit hematopoietic development either autonomously or non-autonomously by interacting with CXCR3 receptors present on HSCs. The role of CXCL10 in inhibiting blood vessel development has been previously reported (Bodnar et al. 2006). During embryonic hematopoiesis, CXCL10 may play a potential role as a feedback signal to attenuate pro-inflammatory signals. Recent work has shown that Interferon gamma (IFN γ) positively regulates AGM hematopoietic development. IP-10 could potentially act as a tuning or feedback system to modulate hematopoietic cluster formation as chronic exposure of IFN γ has been reported to be detrimental for HSC function (Bruin et al. 2015). We have included Figure 3A and 3B as supplementary figures (Supplementary Figure 10) in the manuscript whereas Figure 3C acts as a reference for the reviewer.

Figure 3: Interferon gamma induced protein 10 (IP-10) treatment of CD34+ sorted cord blood cells decreases hematopoietic stem cell (HSC) phenotype. A) Day 7 fold change in induction normalized to control of i) HSC phenotype [CD34+CD90+CD45RA-CD38-/CD34+CD90+CD49f+CD45RA-CD38-], ii) progenitor phenotype [CD34+CD90+CD45RA+/CD34+CD45RA+CD90-/CD34-CD45RA-CD90-] and iii) differentiated phenotype [CD34-]. Bi) Secreted IP-10 concentrations normalized to day 2 with dilution = 0 [D=0] feeding strategy as compared to exponential [exp] feeding strategy, N = 2 biological experiments. ii) Comparison of D=0 and exponential feeding strategies based on total number of cell (TNC), colony forming cell (CFC) and long term culture initiating cell (LTC-IC) assays expansion. C) Meta-analysis of

RNA-seq data displaying CXCR3 and CXCL10 (IP-10) expression from hematopoietic fractions in embryonic day 10.5 AGM tissues containing i) 10-cells or ii) single-cell analysis. . Treatments that share asterisks are significantly different ($p \leq 0.05$). Statistical significance was computed using one-way Student's t-test (Bii and Biii). All error bars represent standard error mean.

5. Transplanted HE cells show relatively low engraftment based on peripheral blood sampling at 10 weeks. The low level of chimerism in the mice should be acknowledged in the text with more discussion about it: the concern is that the in vitro conditions with this system are better than those obtained in vivo, and if the chimerism is not much better than what previous studies obtained using standard cultures, then the utility of this particular culture system becomes questionable.

We thank the reviewer for this question. We have edited the manuscript to summarize the low level of chimerism observed in our experiments. To address the utility of our culture system, we would like to point out that our differentiation platform is serum-free/non-genetically manipulated compared to published platforms (refer to table below edited text). In addition to serum-free conditions, we have established a platform that is robust and efficient in generating PSC-derived hemogenic endothelium cells (~20-fold compared to Ditadi et al, 2015) thereby setting up experimental techniques for future clinical translation.

Our studies also indicate that day 8 HE cells (CD34+VECAD+CD43-CD45-) were capable of short term reconstitution in NSG mice and harbored the majority of blood inducing cells as this fraction had engrafted significantly higher than the negative fraction. Although low levels of mean CD45+ engraftment (0.77%) were observed, these levels fall within the upper range of previously reported engraftment levels by Tian et al. (0.16% - 1.44%) and Wang et al. (<1%). Ledran et al. have reported CD45+ engraftment levels close to 16% however these hPSC-derived cells were cultured on AGM-derived stroma, which is not a clinically relevant system. Higher CD45+ reconstitution levels in primary recipients have been recently reported but in these cases hPSC-derived cells were genetically modified. Low engraftment of hPSC-derived cells may be due to functional differences between embryonic blood progenitors and mature definitive HSCs. For example, the absence of CXCR4 expression on hPSC-derived cells is a requirement for proper homing to the hematopoietic organ. To circumvent this issue, studies have injected hPSC-derived cells intra-femorally into immunocompromised mice as the femur is a more conducive environment for hematopoietic reconstitution (Ran et al. 2013). Interestingly, our studies showed undetectable reconstitution levels in bone marrow samples compared to levels observed in peripheral blood corroborating studies, indicating that embryonic-like blood progenitors lack homing capabilities or may home towards secondary hematopoietic organs in adult recipient mice

Literature Review of Published Studies

Authors/Journal/Year	Total cells injected per mouse recipient	Cell phenotype	Analysis timepoint (weeks)	Highest human engraftment levels	Growth conditions
Wang et al., J. Exp Medicine, 2005	40,000 - 150,000	CD45+CD34+CD38-Lin-	8	1% [Bone Marrow]	Serum-based
Ledran et al., Cell Stem Cell, 2008	500,000	Unsorted	8 - 12	2.86% [Femur]/ 16% [Peripheral]	Stroma-based

				blood]	
Ran et al., Blood, 2013	150,000 - 300,000	CD34+CD45+	9	8% [Femur]	Enforced expression
Average	228000		9.25		

Minor Points

6. The manuscript needs an English language edit in parts of the results and discussion sections (such as lines 204, 291, 298, 374, 443). Please confirm whether a 0.4 µm cell strainer was used (line 402) or correct the number as appropriate. The acronym mESC (line 77) is also not defined.

We have edited the original manuscript to reflect the following:

Line 204: We next tested gene expression levels of hematopoietic-associated factors between blood cells generated from 150-500 and 400-500 micropatterned colonies.

Line 290/291: Mechanistically, we link the p38 MAPK pathway to the decrease of CD45+ blood generation via IP-10 supplementation.

Line: 296-299: The optimized aggregate size for induction to definitive HE phenotypes was 500 cells/aggregate, as this aggregate size yielded 19-fold higher expression of definitive hemogenic endothelium phenotype compared to current protocols (Ditadi & Sturgeon 2015). HE cells possessed multi-lineage potential as they consistently gave rise to myeloid and lymphoid lineages (CD5+CD7+ pro-T cells) - cardinal features of definitive hematopoiesis.

Line 372-374: For hemogenic endothelium induction (10 days), cells were cultured in medium comprised of BMP4 (40 ng/ml, R&D), VEGF (50 ng/ml, R&D), SCF (40 ng/ml, R&D), and bFGF (5 ng/ml, Peprotech). For hematopoietic induction (10 to 25 days), cells were cultured in hematopoietic-inducing medium comprised of SCF (50 ng/ml), VEGF (50 ng/ml), IL-3 (30 ng/ml, R&D), IL-6 (30 ng/ml, R&D), Epo (3 U/ml, R&D), Tpo (30 ng/ml, R&D) and bFGF (5 ng/ml, R&D).

Line 443: MAPK (Thr180/Tyr182, 1:100, Cell Signalling). Secondary staining was carried out using Alexa Fluor

Line 402: through a 40 µm cell strainer, centrifuged and resuspended in StemPro-34 base medium with

Line 77: mESC (mouse embryonic stem cell) colony size manipulation can control JAK-STAT activation, enabling subsequent transition towards

7. Please label the colors in Figure 2B in the images or figure legend.

We have edited figure 2B legend to the following:

(Figure 2B) Immunocytochemistry of day 8 CD34/VECAD fractions show Dil-acetylated-low density lipoprotein (red speckled) and von Willebrand factor production (punctate red) only in positive cell fractions. HUVECs were used as positive controls. Positive fractions and HUVEC cells are concomitantly stained with VECAD (green - VECAD) and Hoechst (blue - Hoechst). Scalebar = 50 µm

8. A reference seems to be lacking regarding IP-10 signaling through p38 MAPK in line 235 (even though some relevant references are found later in the discussion).

We have added the proper reference to line 234:

Line 234: IP-10 has been reported to signal through the p38 mitogen-activated protein kinase (p38 MAPK) pathway⁵⁰

9. The observation that IP-10 serves inhibitory roles is intriguing given that it is secreted in response to interferon- γ (IFN- γ). However, studies have suggested that inflammatory pathways mediated by IFN- γ are important for hematopoietic stem cell emergence in the AGM (work from the Daley, Speck, and Stainier labs). This warrants a brief discussion in text of the proposed inhibitory mechanism of IP-10 and how treatment with IP-10 would be biologically relevant if IFN- γ has an enhancing effect.

We thank the reviewer for this question. We ask the reviewer to refer to point 4 for a brief discussion for the proposed inhibitory mechanism of IP-10.

10. Please provide the total number of events captured or live cells analyzed in the FACS plots in Figure S5 (either in the figure legend or on the plots). I appreciate that the plots for all three recipients were provided.

We have edited Figure S5 legend to the following:

Figure S5: Flow cytometry plots for detecting human engrafted cells in peripheral blood of NSG mice recipients 10 weeks post injection. **(A) CD45+HLA+** (Live cells analyzed: Isotype negative: 454, Replicate #1 negative: 193, Replicate #2 negative: 265, Replicate #3 negative: 155; Isotype positive: 216, Replicate #1: 170, Replicate #2: 217, Replicate #3: 105). **(B) CD45+CD3+** (Live cells analyzed: Isotype negative: 472, Replicate #1 negative: 326, Replicate #2 negative: 93, Replicate #3 negative: 105; Isotype positive: 442, Replicate #1: 122, Replicate #2: 326, Replicate #3: 79) and **(C) CD45+CD41+** (Live cells analyzed: Isotype negative: 210, Replicate #1 negative: 96, Replicate #2 negative: 105, Replicate #3 negative: 35; Isotype positive: 202, Replicate #1: 82, Replicate #2: 293, Replicate #3: 382). Data set is across all biological replicates. Related to Figure 2.

11. Regarding the transplanted HE cells, were the cells frozen and thawed because limited numbers of cells were collected and required multiple rounds for pooling? The freezing process may affect the function of the cells in vivo. Do the transplanted cells persist long-term at 16 weeks and what is the associated long-term chimerism? Is engraftment also present in the bone marrow of recipient mice? That would confirm proper homing mechanisms.

The reviewer is correct. Transplanted HE cells were frozen, thawed and pooled at the time of injection due to the limited number of sorted cells derived from human pluripotent sources. As highlighted above, previous studies that have injected blood progenitor cells derived from hPSC-sources have assessed engraftment at an average of 10 weeks. Although demonstration of definitive hematopoietic engraftment occurs at 16 weeks post-transplant, we assessed engraftment at 10 weeks post-transplant in order to compare our results directly with published literature at this timepoint. From our results, we only observed human engraftment in peripheral blood tissues as opposed to bone marrow. This result is reflected in our revised manuscript as mentioned above. These results indicate that hPSC-derived blood progenitors lack proper homing antigens, warranting further investigation on the proper maturation cues during the differentiation process.

Reviewer #2 (Remarks to the Author):

In this study Rahman et al. employ cell culture patterning technology to investigate how colony size and density affect the conversion of hemogenic endothelium to hematopoietic progenitors during differentiation of human pluripotent stem cells (hPSC). The main conclusions are that colony size and proximity can affect the frequency of generation of hematopoietic progenitors, implicating the microenvironment (and specifically CXCL10 or IP-10) in control of the specification and differentiation process.

The use of colony micropatterning to regulate endogenous signalling pathways crucial for stem and progenitor cell self-renewal or differentiation is a strong approach to understanding the role of intercellular interactions in these processes. The study shows that progenitor cell yield can be influenced by colony size and patterning, and that these parameters likely impact on the production of cytokines and other extrinsic regulators of hematopoiesis. The culture system does not at this stage provide more than an incremental advance towards the generation of bona fide definitive hematopoietic cells but does nicely describe an alternate platform for mimicking an appropriate environment for the process. The experiments are carefully conducted and clearly presented. There are a number of specific points that the authors need to address to improve clarity and precision, as listed below.

Specific points:

1. P4 I 99 and Fig. 1a-do the authors mean low attachment rather than low cluster-I am not sure what a low cluster plate is

The reviewer is correct. We have edited to the following:

Line 98-99: Aggrewell™ plates to low-attachment six well plates....

2. P5 I122-explain for a general readership the significance of CD73+ CD184+. Did the authors look at CD235 (Sturgeon et al. Nat. Biotech. 32 554,2010)?

We have edited the manuscript to describe the significance of CD73+CD184+ expression. :

CD34+VECAD+ HE cells generated using our protocol was composed of 34.5% ± 3.5% CD73-CD184- cells, a substantially higher frequency than has been previously reported. CD184 and CD73 expression has previously been shown to identify arterial and venous endothelium respectively. Lack of both markers can be used to distinguish hemogenic endothelium from vascular endothelium enriched for multi-lineage progenitors that can efficiently give rise to blood cells of myeloid and lymphoid lineages.

In addition, we have assessed CD235a expression in our differentiation protocols and compared its expression to SB-43152 and CHIR99012 supplemented treatments as carried out by Sturgeon et al., 2014, Nature Biotechnology. The results have been included in the manuscript as the following:

The CD34+ expression frequency observed in SB-43152 treated microwell cultures was substantially higher than previously reported [~60% vs 20%](Sturgeon et al. 2014), a result we attribute to the generation of uniform aggregates of optimal size that are generated in microwells. Furthermore, CD235a (a marker indicating progenitor cells primed for primitive hematopoiesis) was minimally expressed in our differentiation culture compared to SB-43152 and CHIR99012 supplemented treatments. Collectively, our data indicate that day 8 CD34+VECAD+ cells are enriched for HE and can be used to investigate EHT transition and blood development.

Under 'HE cells display multi-lineage differentiation' results section, we have included:

Since SB-431542 and CHIR99012 treatment has been reported to promote definitive hemogenic endothelium, we tested T-cell potential using this combination and compared it to control treatments. Our results indicate that CD5+CD7+ generation from SB/CHIR treatment was less efficient compared to control treatments signifying the robustness of our differentiation protocol to generate multi-lineage hemogenic cells.

Figure 4: Hematopoietic differentiation protocol results in similar A) CD235a expression; N = 3 biological replicates. Error bars represent standard error means and B) pro-T cell generation compared to SB/CHIR treatment; N = 1 biological replicate.

3. P5 I132-VWF staining in Fig2b is feeble relative to the positive control, what does isotype control look like?

We have included the isotype control for the reviewer's reference with cell viability stain. The fluorescent pictures indicate the lack of punctate staining which is prevalent in vWF staining in positive and HUVEC controls.

Figure 5: Isotype staining for vWF staining in i) positive fraction [CD34+VECAD+] and ii) negative fraction [CD34-VECAD-]. (Blue – DAPI). Scalebar = 50 μ m.

4. P6 I146-T cell generation is taken as an indicator of definitive hematopoiesis, but there is evidence (Boiers et al Cell Stem Cell 13: 535) that lymphoid cells emerge in the yolk sac prior to definitive hematopoiesis during development

We thank the reviewer for this question. We have used T-cell lineage differentiation as an indicator for definitive hematopoiesis based on studies carried out by Sturgeon et al., 2014, Nature Biotechnology. However, as the reviewer points out, previous studies have outlined the presence of lymphoid progenitors before the onset of HSC emergence (Boiers et al., 2013; Yoshimoto et al., 2012). To consolidate this literature, the Yoder lab has outlined three waves of blood development during embryonic hematopoiesis (Yoder, 2014, Nature Biotechnology). Wave 1 is restricted to primitive erythrocytes and myeloid cells; wave 2 identifies all committed blood progenitors prior to HSC generation (definitive erythroid/myeloid and lymphoid progenitors) and wave 3 refers only to definitive adult-repopulating HSCs. Using this terminology, we have edited line 146 in the manuscript to the following:

T-cell generation from hPSC-derived cells has been used as an assay for generating hematopoietic progenitors during the second wave of embryonic hematopoiesis (Yoder, 2014, Nature Biotechnology).

5. P6 I158-sentence structure “purity analysis” was not “then injected

We apologize for the grammatical error. We have edited the manuscript to the following:

Line 157-159: After thawing, purity analyses of pre-transplanted positive and negative fractions were conducted by flow cytometry (Figure S4B). These fractions were subsequently injected intravenously into sub-lethally irradiated mice.

6. P7 I64- CD41+ percentages do not appear to differ significantly between the two groups in S4C.

We agree with the reviewer. In supplementary figure 4C (now supplementary figure 5C), only CD45+HLA+ and CD45+CD3+ phenotypes between positive and negative fractions are significantly different. Thus these groups are designated with an asterisk whereas CD45+CD41+ (indicating human myeloid engraftment) are similar between positive and negative fractions.

7. P7 I172- what was the composition of the ECM used, and what was the basis for the choice.

As outlined in the methods section (P16, line 397), the extracellular matrix was comprised of fibronectin-gelatin (0.00125%gelatin/0.002% gelatin in ddH₂O). This mixture was previously published as an appropriate substrate for assessing self-renewal conditions in micropatterned hPSC cultures. To clarify we have edited the methods section (line 396) to include the following:

The wells were washed with ddH₂O twice. Next, 100 μL of an ECM mixture comprising of fibronectin-gelatin (0.00125% fibronectin/0.002% gelatin) in ddH₂O was added to each well and the plate was incubated for 3 hours to allow patterned ECM islands to form. The ECM mixture used was previously published as an appropriate substrate for screening pluripotency regulators on micropatterned hPSCs (Nazareth et al., 2013 and Nazareth et al., 2016). After the incubation, the wells were washed twice with PBS and stored at room temperature until initiating cell culture.

8. P7 I177-182-it might be preferable to refer to data showing that cell attachment was not affected by colony patterning here rather than later on.

We agree with the reviewer and thus have edited the manuscript to reflect the following on P7 line 177:

*Colony size was varied, independent of colony pitch, by increasing colony diameter at a fixed pitch (representative images in **Figure 3B**). Colony pitch was manipulated by maintaining colony diameter and varying distance between colony centers. Colony clustering was controlled by configuring colony size and spacing to keep the total number of cells within a well constant (constant global cell density). Representative images and average colonies per manipulation are summarized in Figure S6A and Figure S6B respectively. To ensure that patterning conditions were not selecting for CD34+VECAD+ cells, analysis was performed at 6 h and 5 days post-seeding. Similar patterned cell densities were observed across all treatments before the onset of CD45+ blood generation (Figure S7). Total patterned cells of increasing colony size (Figure S7Ei) and increasing pitch (Figure S7Eii) displayed similar colony-specific cell densities throughout the assay (t = 6 h to t = 5 days post seeding).*

9. P8 I211-the modest differences in transcript levels for these factors would be more convincing evidence of the importance of local paracrine interaction if the authors had provided any information on protein levels. The significance of these data is not really very clear.

Previous work from (Ditadi et al. 2015) have shown that modest increases in hematopoietic gene expression (2 – 10 fold change) can segregate blood progenitor fractions that possess multi-lineage differentiation. To this end, we assessed blood progenitor cells for gene expression instrumental during definitive hematopoiesis such as notch induction (Bertrand et al. 2010; Bigas & Waskow 2016) and blood vessel formation (Zhou et al. 2016; Shalaby et al. 1997). Our results indicate that blood cells derived from 150 μm colonies may possess better definitive blood cell potential due to significantly higher gene expression in both the Notch and blood vessel formation pathways (we kindly ask the reviewer to see Figure 1A within this document for reference).

10. P8 I213 see comment 7 above

We have moved line 213 after line 177 as suggested by the reviewer. Please refer to point 8.

11. P9 I232-perhaps I have overlooked something but the effects of the AntiCXCR3 antibody are not very impressive, and it is unclear precisely how the data on 400-2000 colonies relate to this

We agree with the reviewer that anti-CXCR3 treatment did not significantly enhance CD45+ hematopoietic induction in 150-500 and 400-500 micropatterned colonies. We speculated that endogenous inhibitors (in addition to IP-10) may prevent further enhancement of CD45+ hematopoietic induction when cultured with anti-CXCR3 antibodies. To investigate this phenomenon, we cultured hemogenic endothelium cells as colonies with 400 μm diameter and 2000 μm pitch (400-2000) to further decrease overall cell density (theoretical well coverage = 3%) compared to 150-500 (theoretical well coverage = 7% and 400-500 (theoretical well coverage = 50%). Our results indicate that CD45+ hematopoietic induction from 400-2000 colonies was significantly better than 150-500 (1.4x) and 400-500 (3.2x) colonies inferring that endogenous inhibitors were further mitigated by culturing HE cells as lower coverage micropatterns.

To clarify this point, we have edited the manuscript on page 9, line 242 to the following:

*To investigate our hypothesis, we cultured HE cells as colonies with 400 μm diameter and 2000 μm pitch (400-2000) to further decrease overall cell density (theoretical well coverage = 3%) compared to 150-500 (theoretical well coverage = 7%) and 400-500 (theoretical well coverage = 50%). Our results indicate that CD45+ hematopoietic induction from 400-2000 colonies was significantly better than 150-500 (1.4x, $p \leq 0.05$, **Figure 3F**) and 400-500 (3.2x $p \leq 0.05$, **Figure 3F**) colonies inferring that endogenous inhibitors were further mitigated by culturing HE cells as lower coverage micropatterns. Together these results identify IP-10 as a patterning size-controlled inhibitory molecule of hPSC-derived blood induction.*

12. P13 I310 and elsewhere-the authors should be careful about how they define definitive potential. This is a vexed area of PSC differentiation and claims must be carefully worded and justified.

We agree with the reviewer's comment. We would kindly ask the reviewer to refer to point 4 as to how we justified the use of definitive potential. To clarify this issue we have edited line 308-310 in the manuscript to indicate multi-lineage potential of generated HE cells:

By controlling culture parameters such as aggregate size and oxygen tension; we present a scalable, serum-free differentiation platform that robustly generates hPSC-derived HE with multi-lineage potential.

13. P13 I317-the CXCL10 knockout apparently does not display hematopoietic defects; is it possible that this represents a difference in gene function between mouse and human?

CXCL10 knockout mice display modulation in T-cell recruitment, trafficking and proliferation (Dufour et al. 2002). To our knowledge, there have been no direct reports of hematopoietic stem cell defects in CXCL10 deficient mice. However it has been reported that inflammatory signals such as Interferon gamma (IFN γ) and tumor necrosis factor α (TNF α) are greatly affected in CXCL10 deficient mice, drastically impacting lymphoid cell circulation. As inflammatory signals such as TNF α and IFN γ are important for murine embryonic hematopoiesis, CXCL10 may play a role in modulating these signals. For example, in CXCL10^{-/-} mice, there was an increase in IFN γ producing CD8+ T cells compared to wild type mice (Groom & Luster 2011). These results suggest that CXCL10 can modulate IFN γ expression – a hypothesis which we have postulated in our manuscript as a mechanism to control hematopoietic development. In addition, CXCL9 has been reported to compensate for the loss of CXCL10 induction implying ligand redundancy in the CXCL/CXCR3 growth factor family (Groom & Luster 2011). Ligand redundancy may also explain how embryonic hematopoiesis is unaffected in IFN γ knockout mice although its role in HSC emergence is deemed crucial. The possibility exists that CXCL10 gene function

may differ between mouse and human, however the precise manner in which CXCL10 signals may be context-dependent and warrants further investigation, particularly as it pertains to hematopoietic induction.

REFERENCES

- Bertrand, J.Y. et al., 2010. Notch signaling distinguishes 2 waves of definitive hematopoiesis in the zebrafish embryo. *Blood*, 115(14), pp.2777–83.
- Bigas, A. & Waskow, C., 2016. Blood stem cells: from beginning to end. *Development*, 143(19), pp.3429–3433.
- Bodnar, R.J., Yates, C.C. & Wells, A., 2006. IP-10 blocks vascular endothelial growth factor-induced endothelial cell motility and tube formation via inhibition of calpain. *Circulation Research*, 98(5), pp.617–625.
- Bruin, A.M. De, Voermans, C. & Nolte, M. a, 2015. Review Article Impact of interferon- γ on hematopoiesis. , 124(16), pp.2479–2487.
- Ditadi, A. et al., 2015. Human definitive haemogenic endothelium and arterial vascular endothelium represent distinct lineages. *Nature Cell Biology*, 17(5), pp.580–591.
- Ditadi, A. & Sturgeon, C.M., 2015. Directed differentiation of definitive hemogenic endothelium and hematopoietic progenitors from human pluripotent stem cells. *Methods*, pp.4–11.
- Dufour, J.H. et al., 2002. IFN- γ -Inducible Protein 10 (IP-10; CXCL10)-Deficient Mice Reveal a Role for IP-10 in Effector T Cell Generation and Trafficking. *The Journal of Immunology*, 168(7), pp.3195–3204.
- Groom, J.R. & Luster, A.D., 2011. CXCR3 ligands: redundant, collaborative and antagonistic functions. *Immunology and cell biology*, 89(2), pp.207–15.
- Nazareth, E.J.P. et al., 2013. High-throughput fingerprinting of human pluripotent stem cell fate responses and lineage bias. *Nature methods*, 10(12), pp.1225–31.
- Ran, D. et al., 2013. RUNX1a enhances hematopoietic lineage commitment from human embryonic stem cells and inducible pluripotent stem cells. *Blood*, 121(15), pp.2882–90.
- Shalaby, F. et al., 1997. A Requirement for Flk1 in Primitive and Definitive Hematopoiesis and Vasculogenesis. , 89, pp.981–990.
- Sroczyńska, P. et al., 2009. The differential activities of Runx1 promoters define milestones during embryonic hematopoiesis. *Blood*, 114(26), pp.5279–89.
- Sturgeon, C.M. et al., 2014. Wnt signaling controls the specification of definitive and primitive hematopoiesis from human pluripotent stem cells. *Nature biotechnology*, 32(6), pp.554–61.
- Swiers, G., de Bruijn, M. & Speck, N. a, 2010. Hematopoietic stem cell emergence in the conceptus and the role of Runx1. *The International journal of developmental biology*, 54(6-7), pp.1151–63.
- Zhou, F. et al., 2016. Tracing haematopoietic stem cell formation at single-cell resolution. *Nature*, 533(7604), pp.1–17.

Reviewers' comments:

Reviewer #1 (Remarks to the Author):

The article entitled "Engineering the hemogenic niche mitigates endogenous inhibitory signals and enhances pluripotent stem cell-derived definitive blood progenitor generation" describes serum-free differentiation conditions to generate hemogenic endothelial (HE) cells from pluripotent stem cell (PSC) sources, interestingly by manipulation of colony size and spatial orientation to promote CD45+ cells. The HE cells were transplantable and engrafted into NSG mice. This system enabled the identification of interferon gamma induced 10 (IP-10) among a panel of cytokines as an inhibitor of CD45+ blood induction in the cultures. The authors have done a nice job on their response and follow-up experiments. The authors have addressed most concerns satisfactorily and the paper merits publication once the following points are addressed.

1. I appreciate the author's discussion about the biological relevance of IP-10 in their rebuttal letter. However, the feeding strategy in Figure S10B seems to add confounding factors and it is unclear how it directly assesses the role of IP-10 on the CD34+ cord blood cells. The enhanced effects observed in the exponential feeding group could be a result of more media available to the cells, or that more media simply dilutes out many other inhibitors – especially given that the authors acknowledge the "effects of other endogenous inhibitors" (line 264) a few paragraphs earlier. The exact feeding regimen is unclear and should be described in greater detail in the cord blood methods section such as how the exponential feeding took place.

Although the authors make no claims about the role of IP-10 in non-human systems, the role of IP-10 in *in vitro* cultures seems to become less relevant given that anti-CXCR3 (blocking IP-10 function) does not have an enhanced inductive effect (Figure 3F). Hence, the expression of IP-10 and its receptor *in vivo* in the mouse AGM is still missing. The discussion section would thus benefit at least with some mention of the biological datasets and analysis described in the rebuttal letter to suggest a biological role of IP-10/CXCR3.

2. In Figures S1, S3, S9, and S10, why were only one or two biological replicates done (n=1 or 2) instead of three?

3. The manuscript needs an English language edit in parts of the results and discussion sections. Figure 1D does not appear to be referenced in the text. Lines 126-129 need not be in italics. Line 171 should reference Figure 2D (not Figure 3D). The legend of Figure S2Cii should say CD184 (not CD183). The figures referenced in lines 143 and 161 for Figure S3A and S3C respectively are mismatched and the relevant data is missing altogether. The data referenced in line 139 (Figure S2D) shows only the CD34+ population, but the KDR data referenced in the text is missing. Line 264 should refer to Figure 3F (not Figure 3E).

4. Please add the primer sequence of GAPDH used in Table S1.

Reviewer #2 (Remarks to the Author):

In their revision and rebuttal, the authors have provided a careful, thorough and thoughtful set of replies to the critiques of both reviewers. The clarifications and additional data provided in the revised manuscript address the issues that were raised in those critiques.

Reviewers' comments

Reviewer #1 (Remarks to the Author):

The article entitled "Engineering the hemogenic niche mitigates endogenous inhibitory signals and enhances pluripotent stem cell-derived definitive blood progenitor generation" describes serum-free differentiation conditions to generate hemogenic endothelial (HE) cells from pluripotent stem cell (PSC) sources, interestingly by manipulation of colony size and spatial orientation to promote CD45+ cells. The HE cells were transplantable and engrafted into NSG mice. This system enabled the identification of interferon gamma induced 10 (IP-10) among a panel of cytokines as an inhibitor of CD45+ blood induction in the cultures. **The authors have done a nice job on their response and follow-up experiments. The authors have addressed most concerns satisfactorily and the paper merits publication once the following points are addressed.**

We appreciate this reviewers comments and support.

1. I appreciate the author's discussion about the biological relevance of IP-10 in their rebuttal letter. However, the feeding strategy in Figure S10B seems to add confounding factors and it is unclear how it directly assesses the role of IP-10 on the CD34+ cord blood cells. The enhanced effects observed in the exponential feeding group could be a result of more media available to the cells, or that more media simply dilutes out many other inhibitors – especially given that the authors acknowledge the "effects of other endogenous inhibitors" (line 264) a few paragraphs earlier. The exact feeding regimen is unclear and should be described in greater detail in the cord blood methods section such as how the exponential feeding took place.

We agree with the reviewer that media dilution (via exponential feeding) can dilute other inhibitors in the culture media. In this study, exponential feeding was used to match cell growth (and associated secretion) and defined as an increase of 1.73x total culture volume every two days (i.e day 0 = 1 ml, day 2 = 1.73 ml, day 4 = 3 ml etc) whereas the dilution = 0 feeding regime was defined as a constant culture volume of 1ml. This strategy was used in Csaszar et al. (Csaszar et al., 2012) to reduce the concentration of IP-10, and likely other endogenous inhibitors, over 12 days of culture, resulting in greater cell numbers and enhanced hematopoietic outputs. As pointed out by the reviewer this is a global (not specific to IP-10) effect on potential endogenous inhibitors, thus we have decided to omit Figure S10B from the paper. We note that we have demonstrated a direct effect of IP-10 in Figure S10A where IP-10 supplementation to CD34+ cord-blood derived cells significantly decreased the HSC phenotype. This effect can be rescued by anti-CXCR3 antibody or VX-702 (p38 MAPK inhibitor) addition.

2. Although the authors make no claims about the role of IP-10 in non-human systems, the role of IP-10 to in vitro cultures seems to become less relevant given that anti-CXCR3 (blocking IP-10 function) does not have an enhanced inductive effect (Figure 3F). Hence, the expression of IP-10 and its receptor in vivo in the mouse AGM is still missing. The discussion section would thus benefit at least with some mention of the biological datasets and analysis described in the rebuttal letter to suggest a biological role of IP-10/CXCR3.

We appreciate the reviewers comment. We have now included RNA-sequencing analysis of Zhou et al., 2016, Nature (Zhou et al., 2016) as Supplementary Figure 10B to help address this point. To date, the

highest resolution of gene expression data (single-cell and 10-cell data) pertaining to prospective hemogenic cells during murine AGM hematopoiesis was generated by Zhou et al. This data shows differential expression of IP-10 and CXCR3 expression between pre-HSC and endothelial cells illustrating the role of the IP-10/CXCR3 signalling axis during HSC. In addition we added the following text in the discussion section (starting at line 373):

“Interestingly, IP-10 knockout mice are not known to exhibit hematopoietic defects(Dufour et al., 2002). To explore a potential role for IP-10 during murine embryonic hematopoiesis, we mined literature for IP-10 and CXCR3 gene expression in hemogenic cells during EHT in the AGM. RNA-sequencing analysis of single cell and 10-cell sorted hemogenic populations during murine AGM hematopoiesis(Zhou et al., 2016) revealed the highest IP-10 and CXCR3 expression was present in mature HSC subtypes (defined as T2 pre-HSCs) [Supplementary Fig. 10B]. Supporting this dataset, microarray analysis from E 11.5 AGM tissues(Li et al., 2014) and RNA-sequencing analysis from E 10.5 AGM tissues(Solaimani Kartalaei et al., 2015) indicate that CXCR3 expression is enhanced in the HSC compartment compared to endothelial subtypes. Our data supports a hypothesis that IP-10 can inhibit hematopoietic development either autonomously or non-autonomously by interacting with CXCR3 receptors present on HSCs. Inflammatory signals such as interferon gamma and tumor necrosis factor α are greatly affected in IP-10 deficient mice, drastically impacting lymphoid cell circulation(Groom and Luster, 2011). As inflammatory signals are important for murine embryonic hematopoiesis, IP-10 may play a role in modulating these signals. For example in IP-10^{-/-} mice, there was an increase in IFN γ producing CD8⁺ T cells compared to wild type mice(Groom and Luster, 2011). These results suggest IP-10 can modulate IFN γ expression in a feedback mechanism. During embryonic hematopoiesis, IP-10 could attenuate pro-inflammatory signals since chronic exposure to IFN γ has been reported to be detrimental for HSC function(Bruin et al., 2015).

3. In Figures S1, S3, S9, and S10, why were only one or two biological replicates done (n=1 or 2) instead of three?

We thank the reviewer for this comment. We have amended the manuscript with new data and revision in the following manner to address this comment:

Figure S1: We have carried out an additional biological replicate for comparison among all differentiation protocols to achieve n = 3. The dataset has been updated with no change to previous conclusions.

Figure S10: We have performed two additional biological replicates to investigate the effect of IP-10 on cord-blood derived cells. The dataset has been updated with no change to previous conclusions.

Figure S3: We have removed Figures S3B (T-cell induction in SB/CHIR-supplemented treatments) and Figure S3C (T-cell induction of Y2-1 iPSCs) as they were only done at n = 1 biological replicates. The removal of these datasets does not impact the conclusions of the manuscript because T-cell induction is already robustly shown in Figure 2C.

Figure S9: We are confident in the results we obtained from two biological replicates, of our investigation of the effect of IP-10 on micropatterned HE colonies. Each biological replicate harboured 5 technical replicates for each treatment. Technical replicates allowed analysis of at least 30 micropatterned colonies in one well of a 96 well plate (please see Figure S6 for representative colonies in one 96-well). Thus data for each treatment came from at least 300 micropatterned colonies. We have edited the supplementary Figure S9 to read: B) Fold change in induction normalized to control of (i) VECAD+p-p38⁺ expressing cells, N = 2 biological experiments (\geq 300 micropatterned colonies)

4. The manuscript needs an English language edit in parts of the results and discussion sections. Figure 1D does not appear to be referenced in the text. Lines 126-129 need not be in italics. Line 171 should reference Figure 2D (not Figure 3D). The legend of Figure S2Cii should say CD184 (not CD183). The figures referenced in lines 143 and 161 for Figure S3A and S3C respectively are mismatched and the relevant data is missing altogether. The data referenced in line 139 (Figure S2D) shows only the CD34+ population, but the KDR data referenced in the text is missing. Line 264 should refer to Figure 3F (not Figure 3E).

We thank the reviewer for these detailed comments. We have further edited the manuscript for clarity. We also have amended the manuscript to specifically address the following comments:

Q1) Figure 1D does not appear to be referenced in the text

A1) We have added a reference to Figure 1D in the manuscript text (line 133)

Q2) Lines 126-129 need not be in italics

A2) Lines 126-129 have been fixed to be non-italicized.

Q3) Line 171 should reference Figure 2D (not Figure 3D)

A3) Line 171 (now line 164) is now referencing Figure 2D as opposed to Figure 3D.

Q4) The legend of Figure S2Cii should say CD184 (not CD183)

A4) The legend of Figure S2Cii now reads CD184 instead of CD183

Q5) The figures referenced in lines 143 and 161 for Figure S3A and S3C respectively are mismatched and the relevant data is missing altogether.

A5) We apologize for this error. Figure S3 is now matched with the manuscript text. We have taken out Figure S3B and Figure S3C as these were T-cell induction experiments with a biological replicate of 1.

Q6) The data referenced in line 139 (Figure S2D) shows only the CD34+ population, but the KDR data referenced in the text is missing

A6) We have updated the manuscript text in line 139 to reflect only CD34+ and taken out reference to KDR expression

Q7) Line 264 should refer to Figure 3F (not Figure 3E).

A7) We have changed the reference on line 264 to Figure 3F

5. Please add the primer sequence of GAPDH used in Table S1.

We have added the primer sequence of GAPDH used in this study to Table S1.

Reviewer #2 (Remarks to the Author):

In their revision and rebuttal, the authors have provided a careful, thorough and thoughtful set of replies to the critiques of both reviewers. The clarifications and additional data provided in the revised manuscript address the issues that were raised in those critiques.

We thank the reviewer for their comments.

References

- Bruin, A.M. De, Voermans, C., and Nolte, M. a (2015). Review Article Impact of interferon- γ on hematopoiesis. *124*, 2479–2487.
- Csaszar, E., Kirouac, D.C., Yu, M., Wang, W., Qiao, W., Cooke, M.P., Boitano, A.E., Ito, C., and Zandstra, P.W. (2012). Rapid expansion of human hematopoietic stem cells by automated control of inhibitory feedback signaling. *Cell Stem Cell* *10*, 218–229.
- Dufour, J.H., Dziejman, M., Liu, M.T., Leung, J.H., Lane, T.E., and Luster, A.D. (2002). IFN- γ -Inducible Protein 10 (IP-10; CXCL10)-Deficient Mice Reveal a Role for IP-10 in Effector T Cell Generation and Trafficking. *J. Immunol.* *168*, 3195–3204.
- Groom, J.R., and Luster, A.D. (2011). CXCR3 ligands: redundant, collaborative and antagonistic functions. *Immunol. Cell Biol.* *89*, 207–215.
- Li, Y., Esain, V., Teng, L., Xu, J., Kwan, W., Frost, I.M., Yzaguirre, A.D., Cai, X., Cortes, M., Majenburg, M.W., et al. (2014). Inflammatory signaling regulates embryonic hematopoietic stem and progenitor cell production. *Genes Dev.*
- Solaimani Kartalaei, P., Yamada-Inagawa, T., Vink, C.S., de Pater, E., van der Linden, R., Marks-Bluth, J., van der Sloot, A., van den Hout, M., Yokomizo, T., van Schaick-Solernó, M.L., et al. (2015). Whole-transcriptome analysis of endothelial to hematopoietic stem cell transition reveals a requirement for Gpr56 in HSC generation. *J. Exp. Med.* *212*, 93–106.
- Zhou, F., Li, X., Wang, W., Zhu, P., Zhou, J., He, W., Ding, M., Xiong, F., Zheng, X., Li, Z., et al. (2016). Tracing haematopoietic stem cell formation at single-cell resolution. *Nature* *533*, 1–17.